# The myeloid cell-driven transdifferentiation of endothelial cells into pericytes promotes the restoration of BBB function and brain self-repair after stroke

Tingbo Li[1,2,3,4,5]*, Ling Yang[6], Jiaqi Tu[5], Yufan Hao[5], Zhu Zhu[2,3,4,5], Yingjie Xiong[5], Qingzhu Gao[5], Lili Zhou[2,3,4,5], Guanglei Xie[7], Dongdong Zhang[5], Xuzhao Li[5], Yuxiao Jin[5], Yiyi Zhang[2,3,4,5], Bingrui Zhao[2,3,4,5], Nan Li[6], Xi Wang[7], Jie-Min Jia[2,3,4,5]*

[1]College of Life Sciences, Zhejiang University, Hangzhou, China; [2]Key Laboratory of Growth Regulation and Translation Research of Zhejiang Province School of Life Sciences, Westlake University, Hangzhou, China; [3]Laboratory of Neurovascular Biology, Institute of Basic Medical Sciences, Westlake Institute for Advanced Study, Beijing, China; [4]Westlake Laboratory of Life Sciences and Biomedicine, Hangzhou, China; [5]Laboratory of Neurovascular Biology, School of Life Sciences, Westlake University, Hangzhou, China; [6]High-Performance Computing Center, Westlake University, Hangzhou, China; [7]Westlake Genomics and Bioinformatics Lab, Westlake Laboratory of Life Sciences and Biomedicine, Hangzhou, China

*For correspondence:
litingbo@westlake.edu.cn (TL);
jiajiemin@westlake.edu.cn (J-MinJ)

Competing interest: The authors declare that no competing interests exist.

## eLife Assessment

This **important** study aims to understand the role of endothelial cell differentiation into pericytes in the restoration of blood-brain barrier function after ischemic stroke. Identification of pericytes derived from endothelial cells and the involvement of myeloid cell-derived TGFβ1 signaling are **compelling** new findings, but future studies will be needed to validate the origin and nature of these pericytes. The work will be of interest to blood-brain barrier and basic and translational stroke researchers.

**Abstract** Ischemic stroke, one of the leading causes of death in the world, is accompanied by the dysfunction of the blood-brain barrier (BBB), which aggravates neuron damage. However, the mechanisms underlying the restoration of BBB in the chronic stage after stroke remain unclear. Here, pericyte pool alterations and their consequences for BBB integrity and brain recovery were analyzed in the C57BL/6 mice stroke model. Lineage tracing, RNA-seq, and immunofluorescence staining revealed endothelial cell (EC) transdifferentiation into pericytes (E-pericytes) in C57BL/6 mice after stroke. E-pericytes depletion by diphtheria toxin A (DTA) aggravated BBB leakage and exacerbated neurological deficits in the MCAO model. The myeloid cell-driven transdifferentiation of ECs into pericytes accelerated BBB restoration and brain self-repair after stroke via endothelial-mesenchymal transformation (EndoMT). Decreasing the number of E-pericytes by specific knockout of the *Tgfbr2* gene in ECs also aggravated BBB leakage and exacerbated neurological deficits. EC-specific over-expression of the *Tgfbr2* gene promoting E-pericytes transdifferentiation reduced BBB leakage and exerted neuroprotective effects. Deciphering the mechanism by which E-pericytes coordinate post-stroke recovery may reveal a novel therapeutic opportunity.

## Introduction

All vertebrate creatures have evolved strategies for tissue repair after an accident or conflict, and effective tissue repair is critical for survival. The liver is the unique solid organ capable of precisely regulating its mass through regeneration to maintain the liver-to-body mass ratio within the optimal range required for metabolic homeostasis (*Michalopoulos and Bhushan, 2021*; *Michalopoulos and DeFrances, 1997*). Vascular repair is crucial in liver regeneration and skin wound healing because blood vessels can provide nutrients and oxygen to facilitate tissue restoration and functional recovery in the chronic stage (*Singhal et al., 2018*; *Peña and Martin, 2024*; *Park et al., 2017*). The liver and skin demonstrate a powerful ability for tissue self-repair, which benefits from vascular repair. As one of the most important organs, current knowledge and research on the contribution of vascular repair to brain self-repair are inadequate.

Stroke results in large-scale cell death because of ongoing ischemia and hypoxia, which remains a major cause of adult disability and death (*Broughton et al., 2009*; *Gorelick, 2019*; *Feigin et al., 2018*). The definitive spontaneous repairing processes in the brain remain largely unknown after stroke, and the decoding and harnessing of these processes may accelerate brain recovery, which usually takes years. Until recently, such efforts mainly involved protective trophic factors and cell transplants designed to rescue or replace a specific population of neurons, and the results have largely been disappointing in clinical outcomes (*Barker et al., 2018*; *Gill et al., 2003*; *Bartus and Johnson, 2017*). Treatments to reduce neuronal death and limit acute damage are desirable by timely restoring blood vessel reperfusion in the acute phase (*Juttler et al., 2006*; *Li et al., 2024*). No proven medical therapies exist for restoring cerebral blood flow (CBF) in the chronic phase. Unfortunately, the knowledge of brain self-repair by remedying vascular vessels after stroke in the chronic repair phase is still rare.

The vascular hierarchy comprises arteries, arterioles, capillaries, venules, and veins. Nutrient and gas exchange occur primarily in the capillary beds, up to 90% of blood vessel length (*Murrant and Fletcher, 2022*; *Iadecola et al., 2023*). Capillaries are mainly composed of endothelial cells (ECs) and pericytes, which safeguard the functions of capillary beds (*Trimm and Red-Horse, 2023*). ECs and pericytes respond to different reactions and fates after stroke. Pericytes rapidly exhibited apoptosis and death after stroke; nevertheless, ECs tolerated ischemia and hypoxia (*Hall et al., 2014*). When pericytes are lost, the functions of the blood-brain barrier (BBB) and CBF are destroyed. The BBB serves as a selectively permeable interface that precisely regulates substance exchange between the circulation and the central nervous system (CNS), thereby maintaining the specialized microenvironment essential for proper neuronal signaling (*Ding et al., 2023*; *Armulik et al., 2010*). Effective CBF is essential for cell survival in the brain after stroke. The prevention of BBB leakage and the decrease of CBF due to pericyte loss is important for vascular and neurological functions after stroke.

The endothelium is capable of remarkable plasticity. Certain ECs in the embryo undergo hematopoietic transition, giving rise to multi-lineage hematopoietic stem and progenitor cells, which are crucial for developing the blood system (*Yokomizo and Suda, 2024*). ECs can undergo EndoMT, acquiring mesenchymal properties. The process is important in various pathological conditions, such as fibrosis and cancer (*Xu and Kovacic, 2023*). Endocardial ECs are progenitors of pericytes and smooth muscle cells (SMCs) in the murine embryonic heart via EndoMT (*Chen et al., 2016*). Protein C receptor-expressing (Procr+) ECs would give rise to de novo formation of ECs and pericytes in the mammary gland (*Yu et al., 2016*). In situ, is there a self-repair mechanism to replenish the pericytes pool from ECs for protection functions of blood vessels after stroke in the chronic repair phase that is unknown?

In the acute and subacute phases after stroke, myeloid cells were dominant immune cells entering the brain parenchyma (*Jiang and McCullough, 2024*). Most myeloid cells could not occupy the ischemic brain for a long time and faded away by releasing various factors, including pro-inflammatory, anti-inflammatory, and trophic factors (*Beuker et al., 2022*; *Shichita et al., 2023*; *Gliem et al., 2012*; *Sas et al., 2020*). Blocking myeloid cell recruitment using anti-CCR2 antibody and *CCR2* gene knockout mice impaired long-term spontaneous behavioral recovery after stroke via reducing anti-inflammatory macrophages and angiogenesis (*Fang et al., 2018*; *Pedragosa et al., 2020*; *Wattananit et al., 2016*). In mice subjected to distal middle cerebral artery occlusion (dMCAO) (*Doyle et al., 2010*), TGFβ1 was predominantly co-localized with CD68+ activated microglia and macrophages. Importantly, TGFβ1 contributed to angiogenic pathogenesis in human stroke patients (*Krupinski et al., 1996*). At the

same time, TGFβ is the main driver of EndoMT, which is involved in endothelial-to-pericytic transition in the murine embryonic heart and mammary gland (*Cooley et al., 2014*). However, it remains unclear whether myeloid cells can drive endothelial-to-pericytic transition via EndoMT to replenish the pericyte pool and remedy the functions of vascular vessels after stroke in the chronic phase.

Transdifferentiation of ECs into E-pericytes replenishes the pericyte pool, critically contributing to BBB restoration and brain self-repair after stroke. In the MCAO model, ischemic injury triggered endothelial-to-pericytic transition, driven by myeloid cells via the TGFβ1-TGFβR2-EndoMT pathway. The dynamic changes in E-pericyte numbers directly influenced BBB integrity and long-term functional recovery. The previously overlooked mechanism not only restores the pericyte pool but also enhances endogenous brain repair, offering a novel therapeutic avenue for stroke recovery.

## Results

### Changes in the pericyte pool after stroke

To understand endothelial-to-pericytic transition and its regulatory mechanisms post-stroke, initial efforts targeted changes in the pericyte pool. The MCAO model in mice was employed to mimic stroke conditions in patients (*Beuker et al., 2022*). In the stroke model, CD13$^+$ pericytes and CD31$^+$ ECs exhibited TUNEL positivity in brain parenchyma. Moreover, only half of the pericytes survived at reperfusion 2 days (RP2D), while over 80% of ECs survived (*Figure 1A*). The number of CD13$^+$ pericytes on capillary also decreased at RP2D but then increased at RP7D after stroke (*Figure 1B and C*). Flow cytometry also revealed that the number of pericytes significantly decreased, from 100% on the contralateral side to approximately 50% on the ipsilateral side at RP2D, but gradually increased following reperfusion after stroke (*Figure 1D and E*). Overall, there was a 68.1% increase in the pericyte pool from RP2D to RP34D and a 40.5% increase in total pericytes at RP34D (*Figure 1E*).

Given these results, subsequent efforts focused on determining the reason for increased pericyte numbers in the chronic post-stroke period. EdU was injected into the mice at different times to identify proliferating pericytes and ECs (*Figure 1F*), and it was found that cell proliferation occurred mainly within the first 3 days after stroke (*Figure 1G*). Both pericytes and ECs underwent self-proliferation (*Figure 1H*). EdU$^+$ ECs accounted for fewer than 10% of all ECs, and EdU$^+$ pericytes accounted for approximately 12.9% of all pericytes (*Figure 1I*). EdU$^+$ pericytes and ECs ceased proliferating after RP7D (*Figure 1J*). Thus, 12.9% of the pericytes originated from self-proliferation at RP34D, whereas the origin of the other 27.6% of new pericytes remained unknown.

### scRNA-seq was used to explore the fate of ECs after stroke

To explore the origin of the other 27.6% of new pericytes conversion from ECs after stroke, Cdh5CreERT2 (*Payne et al., 2018*) (induced endothelial cells [iECs]);Ai47 or Ai14 mice were used to label ECs in homeostasis (*Figure 2—figure supplement 1A*). First, ECs were specifically labeled in the brain parenchyma of iECs;Ai47 mice. Without tamoxifen, ECs were not labeled with EGFP (*Figure 2—figure supplement 1B and C*). Immunofluorescence staining showed that EGFP signals were restricted to cells expressing EC markers (CD31$^+$, ERG$^+$, GLUT1$^+$, and VE-Cadherin$^+$) within the brain parenchyma (*Figure 2—figure supplement 1D–G*, *Figure 2—figure supplement 1I*). The EGFP signal did not represent pericyte (CD13$^+$, PDGFRβ$^+$, α-SMA$^+$, and NG2$^+$ cells) cells in the brain parenchyma (*Figure 2—figure supplement 1H–J*). Therefore, vascular ECs in the brain parenchyma were specifically labeled by EGFP from the homeostasis of iECs;Ai47 mice.

EGFP$^+$ cells were isolated from the sham group of iECs;Ai47 mice and the MCAO groups at RP7D and RP34D. After sorting by 10X Genomics and quality control (*Figure 2A*, *Figure 2—figure supplement 2A*), 3568 cells were obtained from the sham group, 4147 cells from the MCAO group at RP7D, and 5070 cells from the MCAO group at RP34D. Different markers were used to identify the cell types based on literature (*Vanlandewijck et al., 2018*; *Kalucka et al., 2020*; *Garcia-Bonilla et al., 2024*; *Figure 2B*) and found 10 major cell types: arterial ECs, venous ECs (2 types), capillary ECs, capillary-venous ECs, SMCs, pericytes, fibroblasts, microglia, and ependymal cells (*Figure 2C*).

Cell identities were defined by established markers: *Pecam1* (ECs), *Gkn3* (arterial ECs), *Slc38a5* (venous ECs), *Rgcc* (capillary ECs), *Rar4* (capillary-venous ECs), *Cnn1* (SMCs), *Pdgfrb* (pericytes), *Pdgfra* (fibroblasts), *Iba1* (microglia), and *Dynlrb2* (ependymal cells) (*Figure 2—figure supplement 2B*). Some unexpected cell types, which might come from the leptomeninges, choroid plexus, periventricular

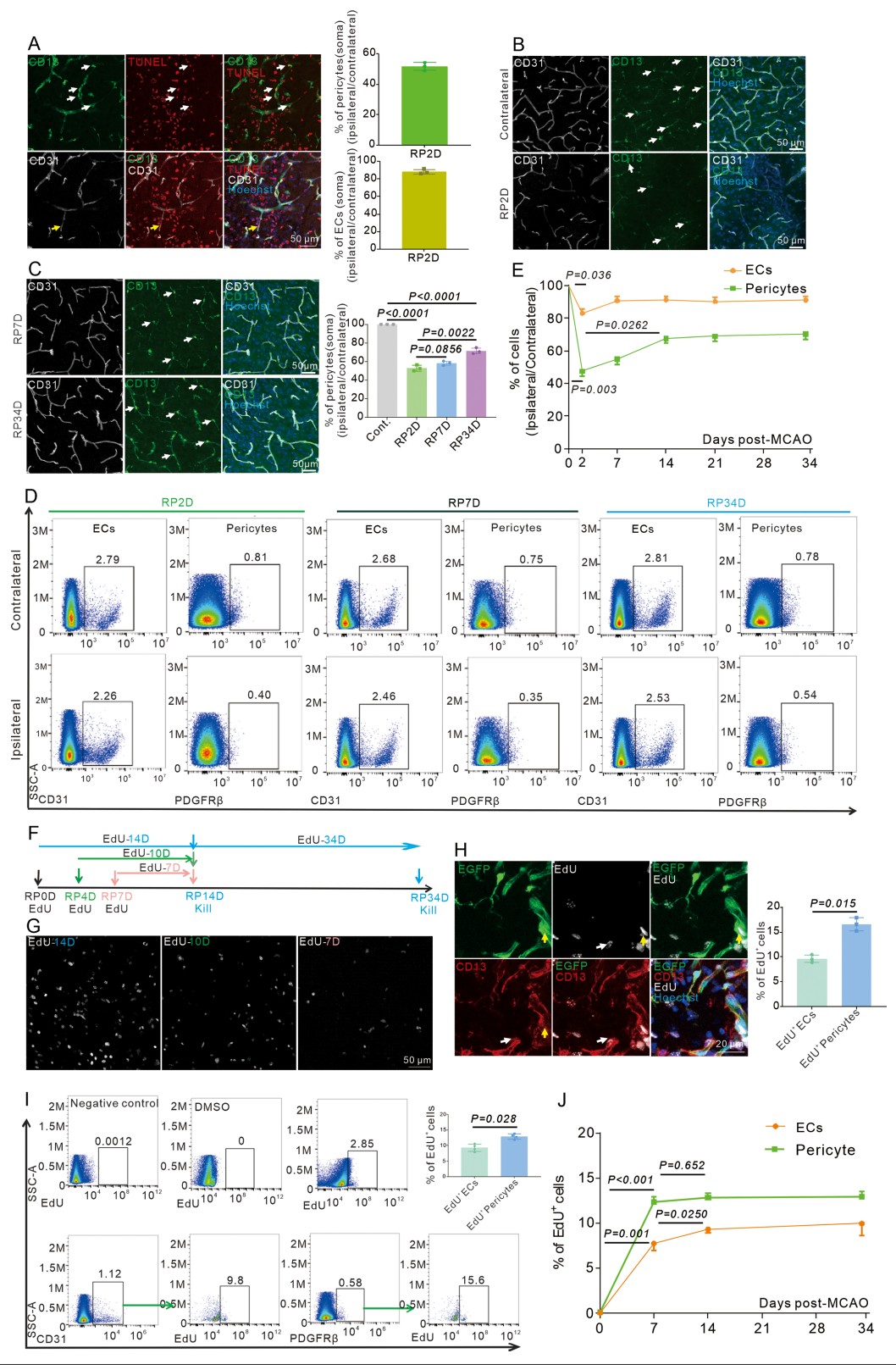

**Figure 1.** Pericytes die rapidly in the acute phase and replenish in the subacute and chronic phases after stroke.
(**A**) Immuflurescence staining shows Tunel+ pericytes (white) and Tunel+ endothelial cells (ECs) (yellow) after
middle cerebral artery occlusion (MCAO) at RP2D and quantifies the proportion of Tunel+ cells in pericytes and
ECs (n = 3, 20 slices/mouse). (**B**) Immunofluorescence staining shows CD13+ soma after MCAO at RP2D (n = 3, 20

*Figure 1 continued on next page*

*Figure 1 continued*

slices/mouse). (**C**) Immunofluorescence staining shows CD13⁺ soma after MCAO at RP7D and RP34D, quantifying the ratio of CD13⁺ soma (n = 3, 20 slices/mouse). (**D**) Flow cytometry analysis of the proportion of pericytes and ECs after MCAO at RP2D, RP7D, and RP34D (n = 6). (**E**) Quantitative analysis of the proportion of pericytes and ECs at different reperfusion times after stroke (n = 6). (**F**) Schematic diagram displaying the time course for EdU injection and analysis time points. (**G**) Maximum EdU signal in the ischemic area at different EdU injection times (n = 3). (**H**) Immunofluorescence staining shows EdU⁺ pericytes (white) and EdU⁺ ECs (yellow) after MCAO at RP34D and quantifies the proportion of EdU⁺ cells in pericytes and ECs (n = 3, 20 slices/mouse). (**I**) Flow cytometry analysis of the proportion of EdU⁺ pericytes and EdU⁺ ECs after MCAO at RP14D (n = 4). (**J**) Flow cytometry analysis of the proportion of EdU⁺ pericytes and EdU⁺ ECs after MCAO at RP7D, RP14, and RP34D (n = 4). Data are presented as mean ± SEM, unpaired two-tailed Student's t-test (C, E, H and J).

cells, peripheral blood in the ischemic area, or contamination during sample processing, were noted. ECs accounted for over 90% of all cells for different groups (*Figure 2—figure supplement 2C*). Gene Ontology (GO) analysis of the top 50 differentially expressed genes (DEGs) (*Figure 2D*) from the 10 major groups showed the functional roles of the genes. GO analysis identified the biological processes, molecular functions, and cellular components associated with the DEGs in each group (*Figure 2D*). The number of cells was calculated in the sham group and MCAO group at RP7D and RP34D, and the proportions of three cell types (pericytes, fibroblasts, and microglia) markedly increased (*Figure 2E*, *Figure 2—figure supplement 2C*).

A heatmap was used to visualize the top DEGs in the pseudotime trajectory (*Figure 2F*) and found that they were related to pericyte transcription factors (*Zhang et al., 2014*) (*Tbx15, Foxc2, Twist1, Tbx18*) and pericyte markers (*Zhang et al., 2014*) (*Cspg4, Pdgfrb, Rgs4*) in cell fate P1 and arterial ECs markers (*Mgp, Gkn3*) in cell fate P2. Among the DEGs, EC-related transcription factors (*Zhang et al., 2014*) (*Erg, Mecom, Foxf2, Foxq1, Sox17, Sox18,* and *Zic3*) were downregulated as differentiation progressed, and the expression of pericyte-related transcription factors (*Tbx15, Foxd1, Tbx18, Heyl,* and *Twist1*) was upregulated, which are critical for pericyte development (*Figure 2G*). Pseudotime trajectory analysis suggested that the developmental trajectory was from ECs to pericytes and arterial ECs (*Figure 2H*). Thus, the scRNA-seq results indicated that ECs might turn into pericyte-like cells after stroke.

## ECs can give rise to pericyte-like cells after stroke

To test whether ECs transdifferentiate into pericyte-like cells post-stroke, lineage tracing in iECs:Ai47 mice was subjected to stroke. The graph showed the time of given tamoxifen, MACO model, and different time points for analysis (*Figure 3—figure supplement 1A*). Laser speckle analysis showed that CBF was reduced by ~80% after MCAO, confirming successful induction of ischemia in mice (*Figure 3—figure supplement 1B*). Immunofluorescence revealed the presence of EGFP⁺ cells in the ischemic area at RP34D. Some cells detached from blood vessels, had long processes with secondary branches, and showed no expression of EC markers (CD31, ERG, GLUT1, and VE-Cadherin) (*Figure 3A*, *Figure 3—figure supplement 1C*). Theoretically, EGFP⁺ cells only existed in ECs and expressed ECs markers in parenchyma at homeostasis. However, the above EGFP⁺ cells lost the characteristics of ECs (EGFP⁺ non-ECs), which indicated those cells had changed fate. Moreover, the EGFP⁺ CD31⁻ cells expressed classic pericyte markers (CD13, PDGFRβ, and NG2) (*Figure 3A–C*), which also indicated ECs might turn into pericyte-like cells after stroke.

Cardiac pericytes upregulated the expression of fibrosis-related genes, exhibiting matrix-synthesizing and matrix-remodeling abilities after myocardial infarction (*Teng et al., 2024*). Some pericytes in the infarct region expressed fibroblast marker PDGFRα, which was beneficial for ECM in myocardial infarction (*Teng et al., 2024*). In the stroke model, EGFP⁺ non-ECs expressed fibroblast marker PDGFRα at RP34D (*Figure 3D*). However, EGFP⁺ non-ECs did not express vimentin (*Figure 3—figure supplement 1D*), which indicates that the EGFP⁺ non-ECs were not fibroblasts, but pericyte-like cells that expressed certain proteins involved in matrix remodeling. Moreover, EGFP⁺ non-ECs did not express microglia markers (Iba1 and CD68) (*Figure 3—figure supplement 1E*), which indicated that EGFP⁺ non-ECs were not microglia.

The proportions of EGFP⁺ non-ECs located on vessels were as follows: 27.8% of total vascular pericytes (*Figure 3E*), 27.5% of total EGFP⁺ non-ECs (*Figure 3F*), 38.1% of migrating EGFP⁺ non-ECs

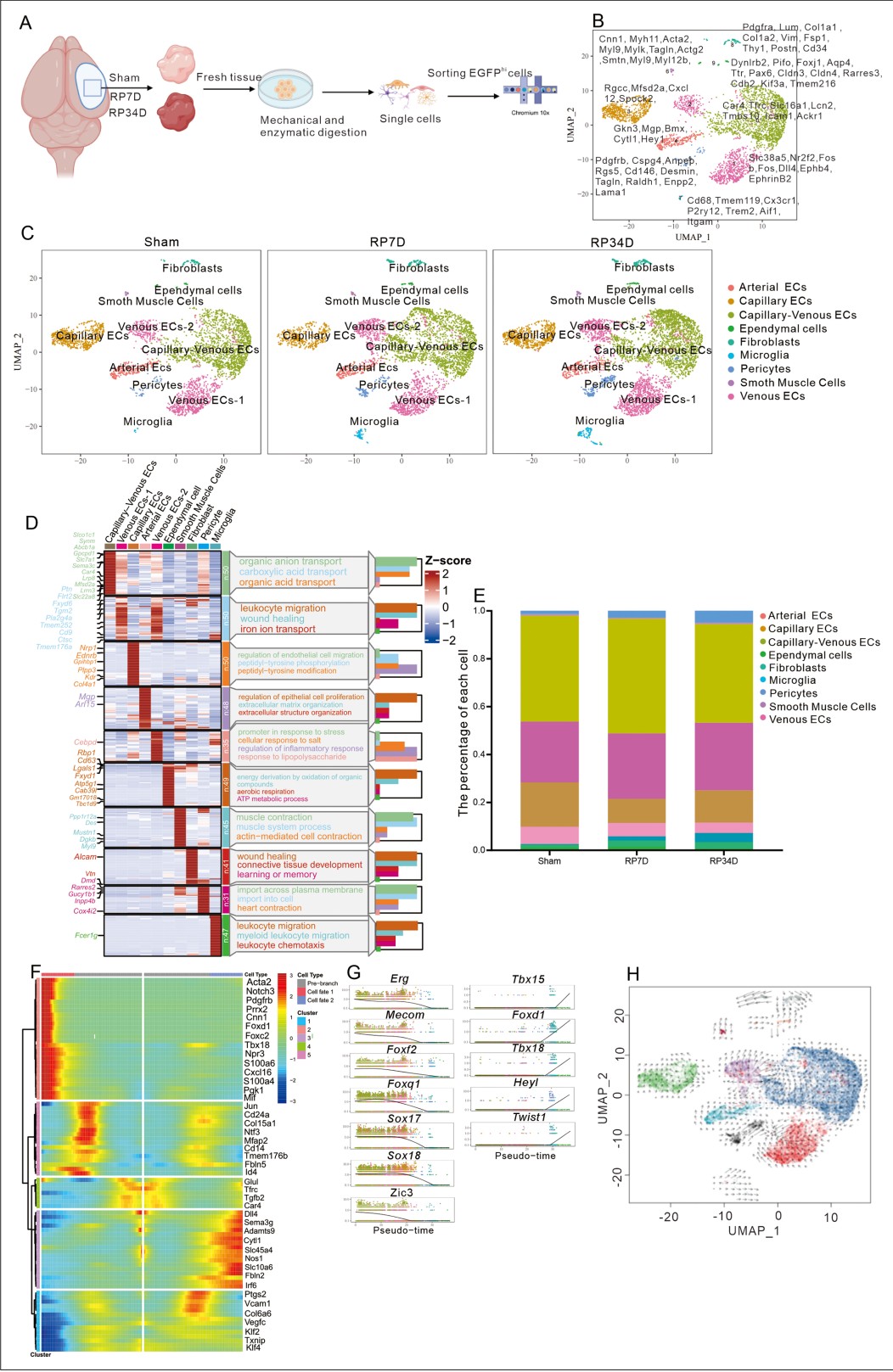

**Figure 2.** scRNA-seq is used to explore the fate of endothelial cells (ECs) after stroke. (**A**) Samples were obtained from mouse ischemic brains at sham, RP7D, and RP34D. Single cells were processed using Chromium 10x 3'DEG chemistry. (**B**) Uniform manifold approximation and projection (UMAP) embedding of all cells and marker genes. (**C**) UMAP analysis of individual sham, RP7D, and RP34D cell transcriptomes showed 10 clusters.

*Figure 2 continued on next page*

*Figure 2 continued*

(**D**) Heatmap displays the expression of the top 50 upregulated genes in each cluster. The scale bar represents the z-score of average gene expression (log). (**E**) Relative proportion of major cell types in different reperfusion times. (**F**) Differential gene expression variance over pseudotime of ECs differentiation trajectory branches. (**G**) Differential transcription factors expression variance over pseudotime of ECs differentiation trajectory branches. (**H**) Pseudotime trajectory of ECs differentiation trajectory branches. The arrows show the direction of pseudotime trajectories.

The online version of this article includes the following figure supplement(s) for figure 2:

**Figure supplement 1.** Cdh5CreERT2 mice specifically recombine parenchymal endothelial cells (ECs).

**Figure supplement 2.** EGFP⁺ cells transcriptomic dataset quality.

(*Figure 3G*), 22.45% of total pericytes (*Figure 3H*). EGFP⁺ non-ECs began to appear at RP7D, and their proportion peaked at RP14D (*Figure 3I*). Furthermore, the EGFP⁺ non-ECs survived over 514 days after MCAO (*Figure 3—figure supplement 1F*). At RP8D, EGFP⁺ non-ECs expressed PDGFRα (*Figure 3J*) and vimentin (*Figure 3K*) but not CD13 (*Figure 3—figure supplement 1G*) and NG2 (*Figure 3— figure supplement 1H*). GO and KEGG analyses also revealed that EGFP⁺ non-ECs expressed genes involved in matrix synthesis, matrix remodeling, and ossification at RP8D (*Figure 3L and M*). EGFP⁺ non-ECs presented the characteristics of fibroblast-like cells at RP8D and pericyte-like cells at RP34D. The majority of the EGFP⁺ non-ECs were located in the penumbra (*Figure 3—figure supplement 1I*), and they were not proliferating, as they were not labeled by EdU at RP34D (*Figure 3—figure supplement 1J*), which indicated that EGFP⁺ non-ECs were converted from ECs via transdifferentiation.

In iECs;Ai47 mice, fewer than 1% of nucleated cells in bone marrow (BM) were labeled with EGFP, main myeloid cells (*Gentek et al., 2018*). Macrophages could invade the brain and develop pericytes in the early phase of vascular development in the CNS (*Prazeres et al., 2018*). Therefore, to eliminate EGFP⁺ macrophages from the BM after stroke, newborn Cdh5CreERT2 P2 mice were intracerebroventricularly injected with AAV2/9-CAG-DIO-EGFP virus. EGFP⁺ non-ECs still expressed the pericyte marker CD13 at RP34D, with the percentage of EGFP⁺ non-ECs expressing CD13 approaching 100% (*Figure 3—figure supplement 1K*). Other methods were also used to specifically label ECs and verify the conversion of ECs to pericytes. In Tie2Dre;Mfsd2aCrexER;Ai47 mice, ECs are specifically labeled in the brain, in which EC could give rise to CD13⁺ EGFP⁺ non-ECs at RP34D (*Figure 3—figure supplement 1L*). Similarly, ECs were specifically infected by AAV2/9-BI30-CAG-EGFP virus in Cdh5CreERT2 mice, and CD13 expression was detected in ~100% of EGFP⁺ non-ECs at RP34D (*Figure 3—figure supplement 1M*). To explore transcriptome characteristics of EGFP⁺ non-ECs at RP34D, EGFP⁺ non-ECs were isolated for RNA-seq analysis (*Figure 3—figure supplement 2A*). Principal component analysis revealed that the cells clustered into ECs and EGFP⁺ non-ECs (*Figure 3—figure supplement 2B*). Analysis of the heatmap showing the top 100 DEGs between ECs and EGFP⁺ non-ECs (*Figure 3—figure supplement 2C*) revealed differences in their transcriptomes. The heatmap showed that EC-specific transcription factor genes (*Zhang et al., 2014*; *Figure 3—figure supplement 2D*), EC-enriched transmembrane receptor genes (*Zhang et al., 2014*; *Figure 3—figure supplement 2E*), and EC-enriched ligand genes (*Zhang et al., 2014*; *Figure 3—figure supplement 2F*) were expressed in ECs but not expressed or expressed at low levels in EGFP⁺ non-ECs. However, pericyte-specific transcription factor genes (*Zhang et al., 2014*; *Figure 3—figure supplement 2G*), pericyte-enriched transmembrane receptor genes (*Zhang et al., 2014*; *Figure 3—figure supplement 2H*), and pericyte-enriched ligand genes (*Zhang et al., 2014*; *Figure 3—figure supplement 2I*) were not expressed or expressed at low levels in ECs but expressed in EGFP⁺ non-ECs. Therefore, EGFP⁺ non-ECs acquired a pericyte-like transcriptome after stroke. Compared with ECs at RP34D, EGFP⁺ non-ECs showed significant upregulation of 379 genes and downregulation of 220 genes (*Figure 3—figure supplement 2J*). GO analysis of upregulated DEGs in EGFP⁺ non-ECs (top 10 GO terms) showed enhanced aerobic capacity (*Figure 3—figure supplement 2K*), whereas ECs exhibited greater dependence on anaerobic metabolism (*De Bock et al., 2013*; *Falkenberg et al., 2019*). KEGG pathway analysis of the upregulated DEGs in EGFP⁺ non-ECs was most significantly enriched in EC migration, epithelial cell migration, epithelial and tissue migration (*Figure 3—figure supplement 2J*), which was consistent with the detachment of EGFP⁺ non-ECs from blood vessels (*Figure 3—figure supplement 1C*). Transcriptomic data indicated pericyte-like features in EGFP⁺ non-ECs, contrasting with

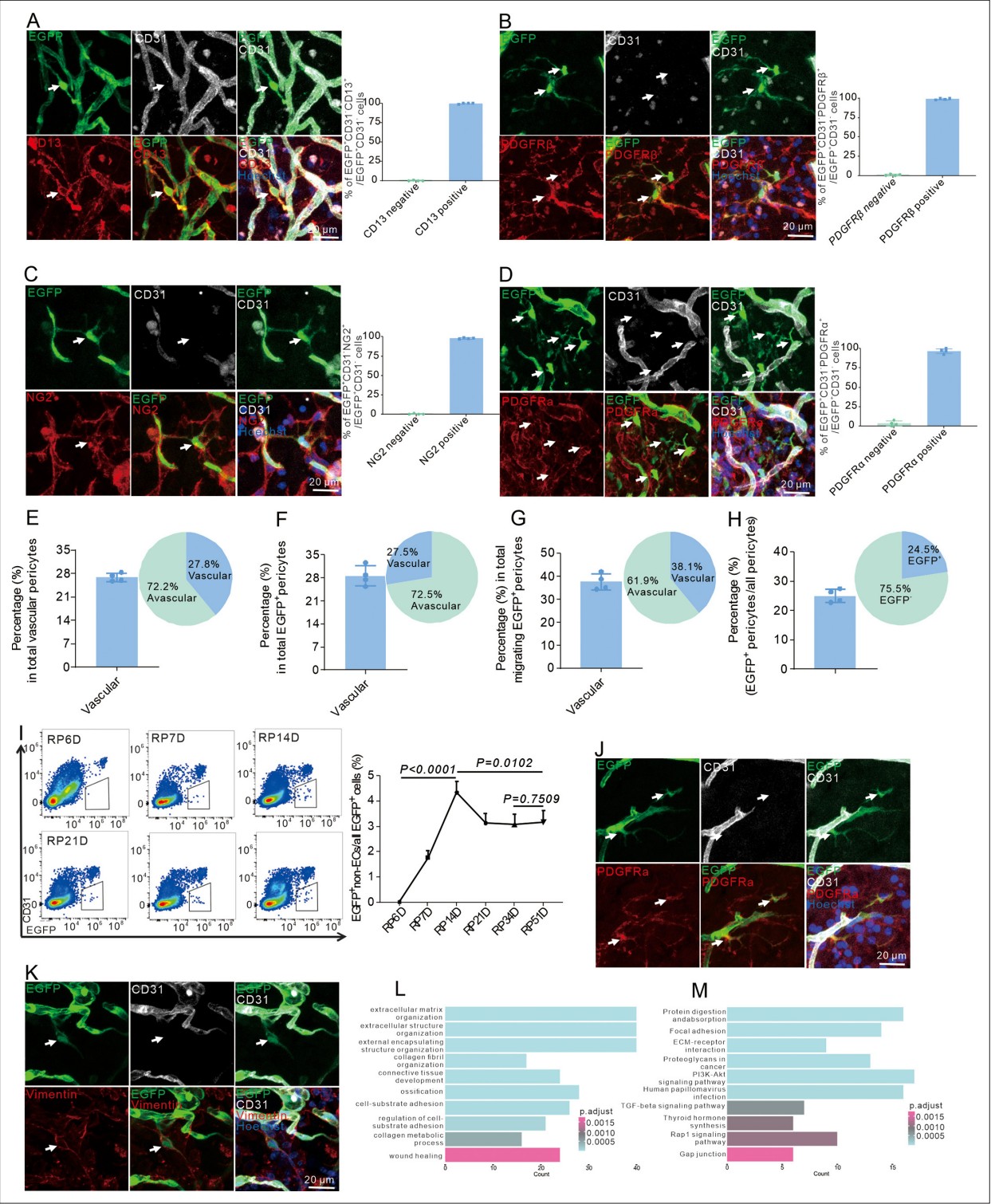

**Figure 3.** Endothelial-to-pericytic transition can replenish pericytes and undergo an intermediate fibroblast-like cell state after stroke. (**A**) Immunofluorescence staining of CD31 and CD13 expression in Cdh5CreERT2;Ai47 mice with middle cerebral artery occlusion (MCAO) at RP34D and quantitative analysis of the proportion of CD13+ cells (n = 5, 20 slices/mouse). (**B**) Immunofluorescence staining of CD31 and PDGFRβ expression in Cdh5CreERT2;Ai47 mice with MCAO at RP34D and quantitative analysis of the proportion of PDGFRβ+ cells (n = 5, 20 slices/mouse). (**C**) Immunofluorescence staining of CD31 and NG2 expression in Cdh5CreERT2;Ai47 mice with MCAO at RP34D and quantitative analysis of the proportion of NG2+ cells (n = 5, 20 slices/mouse). (**D**) Immunofluorescence staining of CD31 and PDGFRα expression in Cdh5CreERT2;Ai47 mice with MCAO at RP34D and quantitative analysis of the proportion of PDGFRα+ cells (n = 5, 20 slices/mouse). (**E**) Quantitative analysis of the proportion of EGFP+

*Figure 3 continued on next page*

*Figure 3 continued*

pericytes on blood vessel/pericytes on blood vessel (n = 5, 20 slices/mouse). (**F**) Quantitative analysis of the proportion of EGFP⁺ pericytes on blood vessel/EGFP⁺ pericytes (n = 5, 20 slices/mouse). (**G**) Quantitative analysis of the proportion of EGFP⁺ pericytes on blood vessels/migrating EGFP⁺ pericytes (n = 5, 20 slices/mouse). (**H**) Quantitative analysis of the proportion of EGFP⁺ pericytes/all pericytes (n = 5, 20 slices/mouse). (**I**) Flow cytometry analysis of the proportion of EGFP⁺&CD31⁻ cells after MCAO at different times and quantitative analysis of the proportion of EGFP⁺&CD31⁻ cells (n = 4). (**J**) Immunofluorescence staining of CD31 and PDGFRα expression in Cdh5CreERT2;Ai47 mice with MCAO at RP8D (n = 5, 20 slices/mouse). (**K**) Immunofluorescence staining of CD31 and vimentin expression in Cdh5CreERT2;Ai47 mice with MCAO at RP8D (n = 5, 20 slices/mouse). (**L**) Gene Ontology (GO) analysis of upregulated differentially expressed genes (DEGs) in EGFP⁺&CD31⁻ cells subgroup, compared with EGFP⁺&CD31⁺ cells from contralateral. (**M**) KEGG pathway analysis of upregulated DEGs in EGFP⁺&CD31⁻ cells subgroup, compared with EGFP⁺&CD31⁺ cells from contralateral. Data are presented as mean ± SEM, unpaired two-tailed Student's t-test (I).

The online version of this article includes the following figure supplement(s) for figure 3:

**Figure supplement 1.** Different means prove that endothelial cells (ECs) can give rise to pericyte-like cells after stroke.

**Figure supplement 2.** Endothelial cells (ECs) give rise to cells with a similar pericyte transcriptome profile after stroke.

immunofluorescence evidence of EC-derived pericyte-like cells post-stroke. Therefore, EGFP⁺ non-ECs are named E-pericytes, as they are derived from ECs.

## Depletion of E-pericytes impedes BBB repair

During both development and adulthood, pericytes maintain BBB function. Reduced pericyte coverage could increase Evans blue accumulation in mutant brain parenchyma (*Armulik et al., 2010*; *Ayloo et al., 2022*). Whether E-pericytes generation also contributes to preserving BBB function after stroke is unknown. To specifically deplete E-pericytes after stroke (*Figure 4A*), AAV2/9-BI30-NG2 promotor-DIO-DeRed (DeRed) virus was first used to label E-pericytes at RP34D (*Figure 4B and C*). AAV2/9-BI30-NG2 promoter-DIO-DTA (DTA) virus, which was injected before MCAO (*Figure 4A*), was used to deplete E-pericytes (*Figure 4D*). Evans blue leakage decreased with increasing reperfusion time, and there was no obvious Evans blue signal at RP34D (*Figure 4E*), which indicated self-repair of the BBB. Depletion of E-pericytes by DTA resulted in a significant increase in Evans blue (*Figure 4F*) and trypan blue leakage (*Figure 4G*) at RP34D after stroke. Furthermore, brain atrophy was more pronounced following the depletion of E-pericytes by DTA at RP34D after stroke (*Figure 4F*). E-pericytes prevent vessel leakage and preserve BBB function after stroke.

## Depletion of E-pericytes aggravates neurological deficits after stroke

Depletion of pericytes resulting in BBB dysfunction can impact normal brain function, damage neurons, and impair cognitive and neurological functions (*Andjelkovic et al., 2023*; *Zhao et al., 2015*). E-pericytes generation after stroke might contribute to neuron survival and facilitate spontaneous behavioral recovery by restoration of BBB function.

Depletion of E-pericytes by DTA resulted in a decreased survival rate (*Figure 5A*), a decreased latency to fall in the rotarod test (*Figure 5B*), an increased frequency of falling from the pole (*Figure 5C*), an increased deviation from normal in the corner test (*Figure 5D*), and an increased difficulty in removing adhesive tape from the forepaw (*Figure 5E*). Quantitative immunofluorescence analysis revealed a reduction in NeuN⁺ neurons after stroke when E-pericytes were depleted by DTA (*Figure 5F*). Furthermore, E-pericytes expressed higher levels of genes related to neuronal survival and growth, compared to ECs (*Figure 5G*). Collectively, E-pericytes promote neuronal survival, leading to spontaneous behavioral recovery after stroke.

Additionally, when E-pericytes were depleted by DTA, there was a decrease in CBF at RP34D after stroke (*Figure 5H*). The number of CD31⁺ ECs and PDGFRβ⁺ pericytes decreased in immunofluorescence at RP34D after stroke when E-pericytes were depleted by DTA (*Figure 5I and J*), indicating decreased microvasculature. When E-pericytes were depleted, the number of CD13⁺ pericytes in the ischemic area was reduced by flow cytometry at RP34D after stroke, and the EC/pericyte ratio was increased (*Figure 5I and J*), suggesting that vascular integrity was disrupted. E-pericytes maintain the EC/pericyte ratio and preserve normal vascular functions.

## The TGFβ-TGFβR2 pathway impacts EndoMT and E-pericytes

Lineage tracing revealed that Procr⁺ ECs could give rise to pericytes during mammary gland development, and the transcriptome of Procr⁺ ECs exhibited the feature of EndoMT (*Yu et al., 2016*).

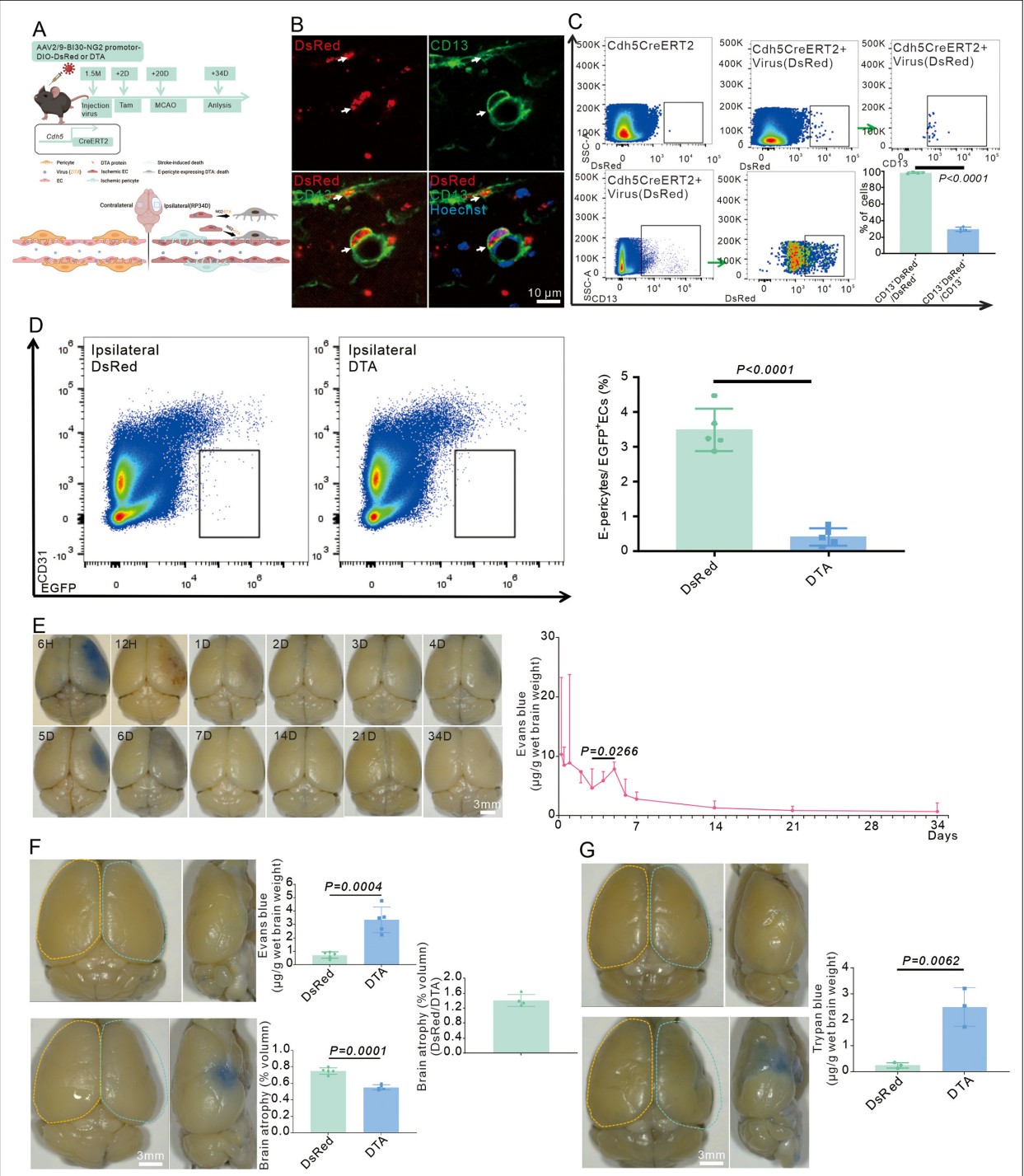

**Figure 4.** E-pericytes deletion by AAV2/9-BI30-DIO-NG2-promotor-DTA virus aggravates blood-brain barrier (BBB) leakage after stroke. (**A**) Schematic diagram displaying Cdh5CreERT2 injected with AAV2/9-BI30-DIO-NG2-promotor -DsRed or DTA virus to kill the cell and the time course for tamoxifen, middle cerebral artery occlusion (MCAO), and analysis time points. (**B**) Immunofluorescence staining of CD13 expression in Cdh5CreERT2 injected with AAV2/9-BI30-DIO-NG2-promotor-DsRed A virus (n = 3). (**C**) Flow cytometry analysis of the proportion of DsRed⁺&CD13⁺/DsRed⁺ cells and quantitative analysis of the proportion (n = 3). (**D**) Flow cytometry analysis of the proportion of E-pericytes cells in Cdh5CreERT2 injected with AAV2/9-BI30-DIO-NG2-promotor-DTA virus and quantitative analysis of the proportion (n = 5). (**E**) Image showing the leakage of Evans blue in wild-type (WT) mice at different times after MCAO and quantitative analysis of the leakage of Evans blue (n = 3). (**F**) Image showing the leakage of Evans blue in Cdh5CreERT2 injected with AAV2/9-BI30-DIO-NG2-promotor-DTA virus at RP34D after MCAO, quantitative analysis of the leakage of Evans blue (n = 5), and the brain atrophy volume (n = 5). (**G**) Image showing the leakage of trypan blue in Cdh5CreERT2 injected with AAV2/9-BI30-DIO-NG2-promotor-DTA virus at RP34D after MCAO and quantitative analysis of the leakage of trypan blue (n = 3). Data are presented as mean ± SEM, unpaired two-tailed Student's t-test (C, D, F and G).

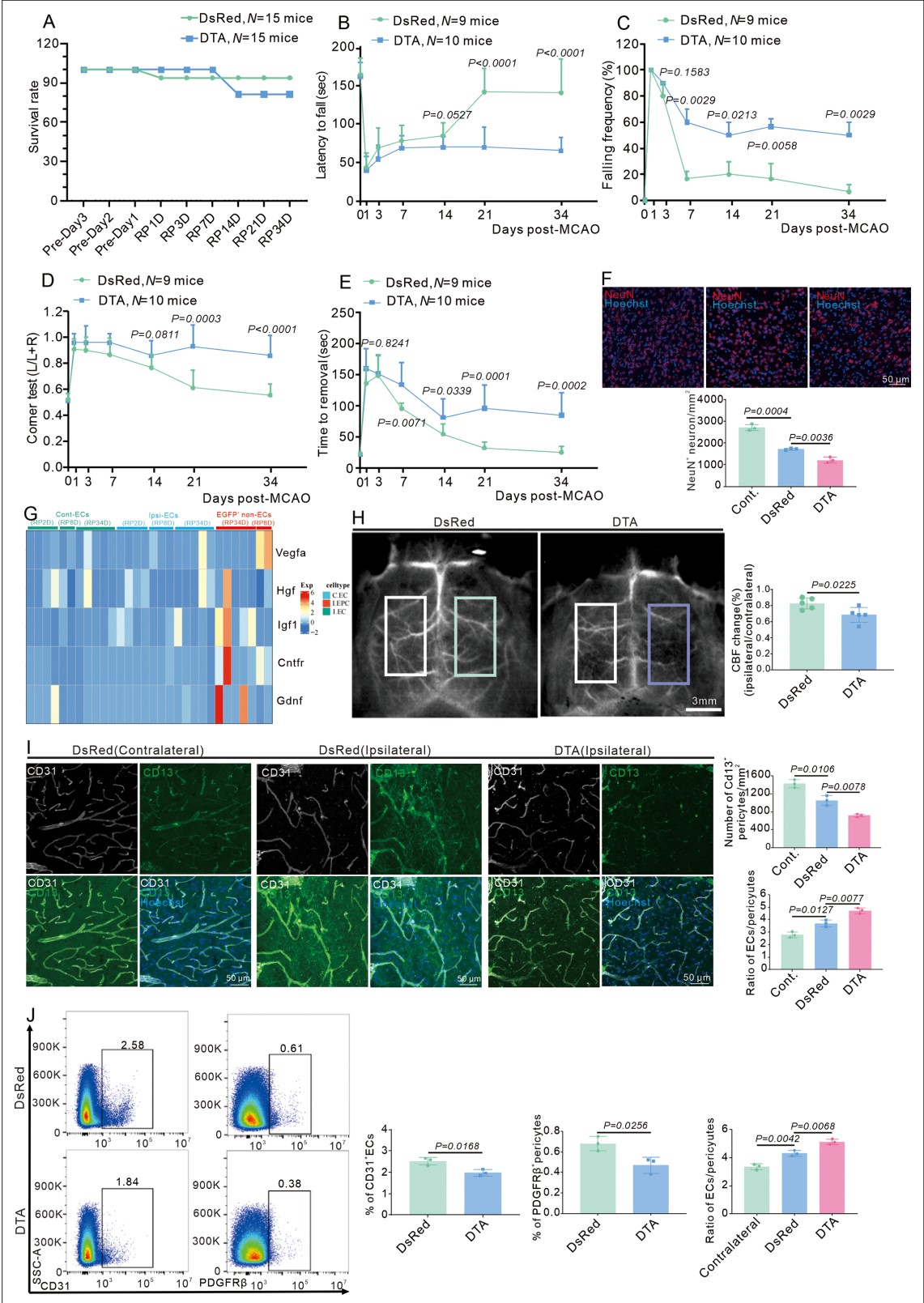

**Figure 5.** E-pericytes deletion by AAV2/9-BI30-DIO-NG2-promotor-DTA virus exacerbates neurological deficit after stroke. (**A**) Graph showing the survival rate of mice in Cdh5CreERT2 mice injected with AAV2/9-BI30-DIO-NG2-promotor-DsRed or DTA virus after middle cerebral artery occlusion (MCAO) at RP34D (n = 15). (**B**) Graph showing rotarod test in Cdh5CreERT2 mice injected with AAV2/9-BI30-DIO-NG2-promotor-DsRed or DTA virus after MCAO (n = 9–10). (**C**) Graph showing beam walking test in Cdh5CreERT2 mice injected with AAV2/9-BI30-DIO-NG2-promotor-DsRed or DTA virus

*Figure 5 continued on next page*

*Figure 5 continued*

after MCAO (n = 9–10). (**D**) Graph showing corner test in Cdh5CreERT2 mice injected with AAV2/9-BI30-DIO-NG2-promotor-DsRed or DTA virus after MCAO (n = 9–10). (**E**) Graph showing adhesive movement test in Cdh5CreERT2 mice injected with AAV2/9-BI30-DIO-NG2-promotor-DsRed or DTA virus after MCAO (n = 9–10). (**F**) Immunofluorescence staining of NeuN expression in Cdh5CreERT2 mice injected with AAV2/9-BI30-DIO-NG2-Long-DsRed or DTA virus after MCAO at RP34D and quantitative analysis of the number in the unit area (n = 4, 20 slices/mouse). (**G**) The heatmap showing promotion neuron survival and growth genes expression in all groups (n = 5). (**H**) Image showing the change of cerebral blood flow (CBF) in Cdh5CreERT2 mice injected with AAV2/9-BI30-DIO-NG2-promotor-DsRed or DTA virus after MCAO at RP34D and quantitative analysis of the change of CBF (n = 5). (**I**) Immunofluorescence staining of CD13 and CD31 expression in Cdh5CreERT2 mice injected with AAV2/9-BI30-DIO-NG2-promotor-DsRed or DTA virus after MCAO at RP34D and quantitative analysis of the percentage (n = 3, 20 slices/mouse). (**J**) Flow cytometry analysis of the proportion of CD13$^+$ cells and CD31$^+$ cells in Cdh5CreERT2 mice injected with AAV2/9-BI30-DIO-NG2-promotor-DsRed or DTA virus after MCAO at RP34D and quantitative analysis of the percentage (n = Immunofluorescence staining of p-SMAD3 expression 3). Data are presented as mean ± SEM, unpaired two-tailed Student's t-test (A-F and H-J).

KEGG analysis revealed that the components of the TGFβ signaling pathway, the main driving force for EndoMT, were highly expressed in EGFP$^+$ non-ECs at RP8D (*Figure 3M*). EndoMT potentially participated in E-pericyte differentiation. Immunofluorescence showed that EndoMT occurred in the brain after stroke. The expression of EndoMT marker p-SMAD3 (*Li et al., 2018*) increased in ECs until RP8D and decreased at RP34D after stroke (*Figure 6A*). At RP8D, two adjacent EGFP$^+$ cells were observed on the same vessel: a CD31$^+$p-SMAD3$^-$ cell exhibiting EC-like characteristics and a CD31$^-$p-SMAD3$^+$ EGFP$^+$ non-EC, indicating active EndoMT (*Figure 6B*). Moreover, the expression of EndoMT markers KLF4 and α-SMA (*Li et al., 2018*) increased in ECs until RP8D and decreased at RP34D after stroke (*Figure 6—figure supplement 1A and B*). The expression of α-SMA in ECs increased, as shown by flow cytometry analysis of cytoplasmic proteins (*Figure 6—figure supplement 1C*). EndoMT marker genes are also expressed in ECs on the ipsilateral side at RP2D (*Figure 6—figure supplement 1D*). Immunofluorescence and flow cytometry revealed that ECs underwent EndoMT after stroke. Flow cytometry analysis of cytoplasmic proteins revealed that 1.6% of CD31$^+$ ECs showed α-SMA expression at RP2 (*Figure 6—figure supplement 1C*). The percentage was comparable to the ratio of E-pericytes to CD31$^+$ ECs. Treatment with TGFβR2 inhibitors, L6293 and S1067, reduced the expression of EndoMT markers (*Figure 6—figure supplement 1E*), and ECs-specific knockout of the *Tgfbr2* gene also decreased the expression of EndoMT markers (*Figure 6C*, *Figure 6—figure supplement 1F*). The TGFβR2 inhibitors diminished the number of E-pericytes (*Figure 6—figure supplement 1G and H*), and similarly, EC-specific knockout of the *Tgfbr2* gene diminished the proportion of E-pericytes (*Figure 6D and E*). Infection with AAV2/9-BI30-EF1α-DIO-Tgfbr2-3XFLAG-P2A-DsRed-WPREs[Virus(Tgfbr2)] resulted in EC-specific overexpression of the Tgfbr2 protein (*Figure 6—figure supplement 1I*), increased the expression of EndoMT markers (*Figure 6F*, *Figure 6—figure supplement 1J*) and the generation of E-pericytes (*Figure 6G and H*). Stroke activates TGFβ-TGFβR2 signaling and enhances E-pericytes, indicating that the endothelial-to-pericytic transition occurs via the TGFβ-TGFβR2-EndoMT pathway.

## Infiltrating myeloid cells express TGFβ1 in the brain after stroke

The TGFβ-TGFβR2 pathway is involved in EndoMT and the generation of E-pericytes after stroke. Therefore, the source of TGFβ was explored after stroke. The three main TGFβ isoforms in mammals are TGFβ1, TGFβ2, and TGFβ3, and TGFβ1 accounts for more than 90% of total TGFβ (*Jenkins, 2008*). In young female mice subjected to dMCAO, the TGFβ1 protein predominantly co-localized with CD68, activated microglia and macrophages, at 3 days post-stroke (*Doyle et al., 2010*). RNAscope analysis revealed that *Tgfb1* mRNA was present predominantly (over 85%) in CD45$^+$ immune cells at RP1D (*Figure 7A*) and RP2D (*Figure 7—figure supplement 1A*) but was present in only a fraction of Iba1$^+$ microglia (~15%) (*Figure 7—figure supplement 1B*). Moreover, *Tgfb1* mRNA was not expressed in CD13$^+$ pericytes (*Figure 7—figure supplement 1C*).

During the acute and subacute phases after stroke, immune cells that infiltrate the ischemic brain are primarily of the myeloid lineage (*Jiang and McCullough, 2024*). Initially, flow cytometric analysis revealed that CD45$^{hi}$ immune cells were predominantly of the myeloid lineage after stroke, with the majority being monocytes (*Figure 7—figure supplement 1D–F*). CD45$^+$ cells were sorted 2 days after MCAO, and single-cell sequencing analysis revealed that these cells were also primarily of the myeloid lineage (*Figure 7B*). Among myeloid cells, monocyte-derived macrophages (MDMs) were the main cells expressing *Tgfb1* mRNA (*Figure 7C*), with 96.3% of *Tgfb1* transcripts detected in this population

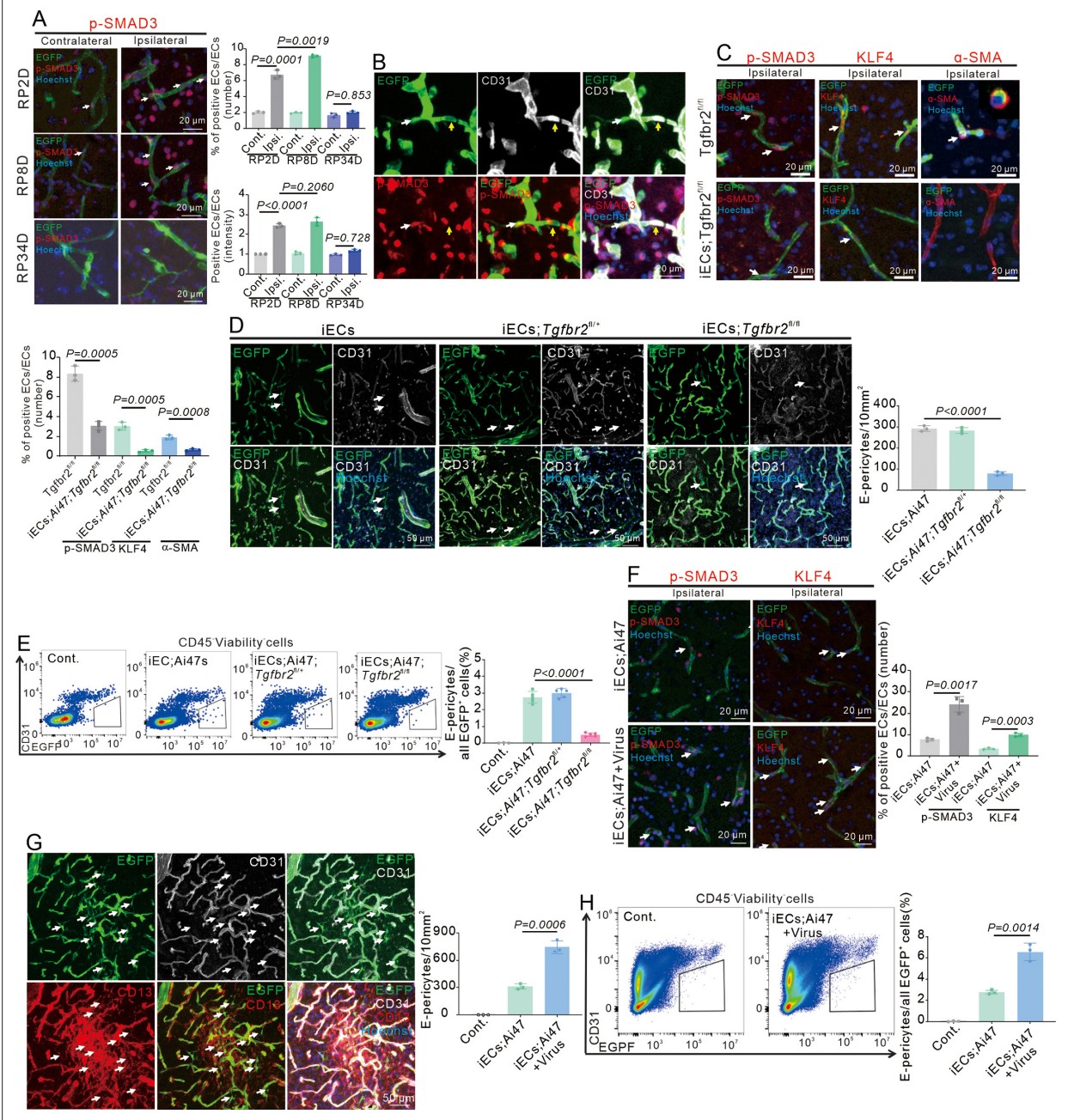

**Figure 6.** Endothelial-specific loss and reinforcement of *Tgfbr2* gene expression affect endothelial-mesenchymal transformation (EndoMT) and E-pericytes. (**A**) Immunofluorescence staining of p-SMAD3 expression in Cdh5CreERT2;Ai47 mice at RP2D, RP8D, and RP34D and quantitative analysis of the proportion and intensity (n = 3, 20 slices/mouse). (**B**) Immunofluorescence staining of CD31 and p-SMAD3 expression in Cdh5CreERT2;Ai47 mice at RP8D (white:EGFP⁺&p-SMAD3⁺&CD31⁻; yellow:EGFP⁺&p-SMAD3⁻&CD31⁺; n = 3). (**C**) Immunofluorescence staining of EndoMT markers (p-SMAD3, KLF4, and α-SMA) expression in Cdh5CreERT2;Ai47;Tgfbr2^fl/fl mice at RP2D and quantitative analysis of the proportion and intensity (n = 3, 20 slices/mouse). (**D**) Immunofluorescence staining of CD31 expression in Cdh5CreERT2;Ai47;Tgfbr2^fl/fl mice at RP34D and quantitative analysis of the number in the unit area (n = 3, 20 slices/mouse). (**E**) Flow cytometry analysis of the proportion of E-pericytes in Cdh5CreERT2;Ai47;Tgfbr2^fl/fl mice at RP34D and quantitative analysis of the proportion (n = 5). (**F**) Immunofluorescence staining of EndoMT markers (p-SMAD3⁺ and KLF4⁺) expression in Cdh5CreERT2;Ai47 mice injected with AAV2/9-BI30-EF1α-DIO-Tgfbr2-3XFLAG-P2A-DsRed-WPREs at RP2D and quantitative analysis of the proportion of EndoMT markers (n = 3, 20 slices/mouse). (**G**) Immunofluorescence staining of CD31 and CD13 expression in Cdh5CreERT2;Ai47 mice with AAV2/9-BI30-EF1α-DIO-Tgfbr2-3XFLAG-P2A-DsRed-WPREs at RP34D and quantitative analysis of the number in unit area (n = 3, 20 slices/mouse). (**H**) Flow cytometry analysis of the proportion of E-pericytes in Cdh5Cre ERT2;Ai47 mice with AAV2/9-BI30-EF1α-DIO-Tgfbr2-3XFLAG-P2A-DsRed-WPREs at RP34D and quantitative analysis of the proportion of E-pericytes (n = 3). Data are presented as mean ± SEM, unpaired two-tailed Student's t-test (A and C-H).

*Figure 6 continued on next page*

*Figure 6 continued*

The online version of this article includes the following figure supplement(s) for figure 6:

**Figure supplement 1.** Systemic inhibition of TGFβR2 reduces endothelial-mesenchymal transformation (EndoMT) and E-pericytes.

(*Figure 7D*). Immunofluorescence staining revealed almost no protein expression of TGFβ1 on the contralateral side (*Figure 7E*) but showed the presence of the TGFβ1 protein at RP2D. However, TGFβ1 protein expression was no longer observed at RP8D (*Figure 7F*). Furthermore, after anti-Ly6C/Ly6G antibodies were used to eliminate myeloid cells (*Figure 7—figure supplement 1G and H*), *Tgfb1* mRNA levels (*Figure 7G*) and Tgfb1 protein expression (*Figure 7H*) decreased. Myeloid cells could infiltrate the brain and release dominantly TGFβ1 after stroke.

## Infiltrating myeloid cells drive the generation of E-pericytes, promote BBB recovery and neurological recovery after stroke

RNAscope showed that myeloid cells could infiltrate the brain and release dominantly TGFβ1. The role of myeloid cells in driving E-pericyte-mediated BBB repair and neurological recovery was examined in a post-stroke model. First, the administration of anti-Ly6C/Ly6G antibodies to eliminate myeloid cells significantly reduced the number of CD45$^{hi}$ immune cells in the brain at RP34D (*Figure 8—figure supplement 1A and B*), and it also decreased the generation of E-pericytes (*Figure 8A and B*). Furthermore, there was a significant increase in Evans blue extravasation and brain atrophy at RP34D after stroke when infiltrating myeloid cells were eliminated (*Figure 8C*). The inhibition of infiltrating myeloid cell-driven E-pericytes had affected BBB function after stroke. Elimination of infiltrated myeloid cells at RP34D led to reduced CBF after stroke (*Figure 8—figure supplement 1C*). The number of CD31$^+$ ECs was quantified and found that it decreased when infiltrating myeloid cells were eliminated (*Figure 8—figure supplement 1D*), indicating decreased ECs-mediated angiogenesis. The number of CD13$^+$ pericytes in the ischemic area decreased (*Figure 8—figure supplement 1D*). Subsequent elimination of infiltrating myeloid cells further increased the EC/pericyte ratio (*Figure 8—figure supplement 1D and E*), indicating disrupted vascular integrity.

Immunofluorescence statistics revealed a reduction of NeuN$^+$ neurons after ischemic stroke when infiltrating myeloid cells were eliminated (*Figure 8D*), indicating that fewer neurons survived. The elimination of infiltrating myeloid cells resulted in a low survival rate (*Figure 8E*). Moreover, the elimination of infiltrating myeloid cells decreased the latency to fall in the rotarod test (*Figure 8F*), increased the frequency of falling from the pole (*Figure 8G*), increased the deviation from normal in the corner test (*Figure 8H*), and impaired the removal of adhesive tape from the forepaw (*Figure 8I*). Myeloid cells infiltrate the brain and release TGFβ1, inducing the formation of E-pericytes after stroke. E-pericytes sustain vascular integrity by supporting both ECs and pericytes, thereby protecting vascular function and promoting neuronal survival.

## EC-specific knockout of the *Tgfbr2* gene aggravates BBB leakage and neurological deficits after stroke

Given that infiltrating myeloid cells promote EC-to-pericyte transdifferentiation via TGFβ1-TGFβR2 signaling, EC-specific blockade of the pathway could further affect BBB leakage and neurological deficits through E-pericyte regulation. To this end, EC-specific overexpression of the *Tgfbr2* gene increased the proportion of E-pericytes and EC-specific expression of the *DTA* gene in transformed cells to eliminate E-pericytes. When the *Tgfbr2* gene was overexpressed in ECs to increase E-pericytes and DTA was expressed in transformed cells to deplete E-pericytes, there was no significant change in the number of E-pericytes in the Tgfbr2+DTA group compared with the DTA group (*Figure 9A*). Moreover, there was no difference in BBB leakage between the *DTA* expression group and the group in which *Tgfbr2* was overexpressed and *DTA* was expressed (*Figure 9B*). Evans blue leakage increased at different time points after stroke in mice with EC-specific loss of the *Tgfbr2* gene, even up to RP118D after stroke (*Figure 9C and D*). Moreover, trypan blue leakage increased at RP34D with EC-specific loss of the *Tgfbr2* gene (*Figure 9—figure supplement 1A*). BBB leakage did not change significantly from RP7D to RP14D, while the proportion of E-pericytes increased. However, BBB leakage increased significantly after the elimination of E-pericytes (*Figure 9D*), which indicated that E-pericytes promoted the repair of BBB from RP7D to RP14D. Interestingly, no obvious

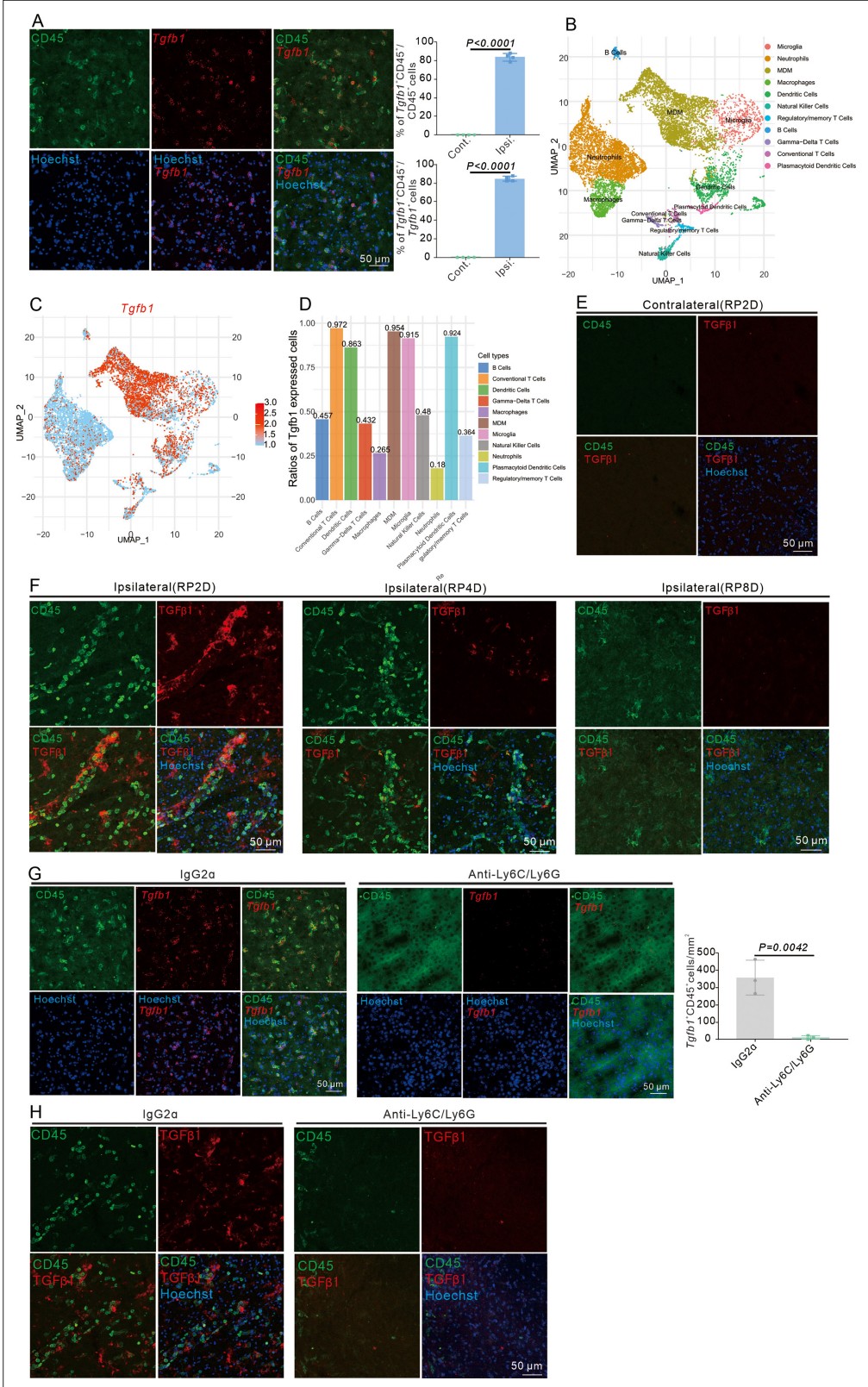

**Figure 7.** Infiltrating myeloid cells produce dominant TGFβ1 in mouse brain after stroke. (**A**) Immunofluorescence staining of CD45 and *Tgfb1* gene expression in wild-type (WT) mice with middle cerebral artery occlusion (MCAO) at RP2D and quantitative analysis of the proportion of *Tgfb1+* cells (n = 4, 10 slices/mouse). (**B**) scRNA-seq of CD45+ cells isolated from the ipsilateral brain with MCAO:2H and RP1.5D. Dimensionality reduction and

*Figure 7 continued on next page*

*Figure 7 continued*

identification of clusters of transcriptionally similar cells were performed in an unsupervised manner (n = 1), monocyte-derived macrophage (MDM). (**C**) *Tgfb1* gene expression in dimensionality reduction and identification of clusters (n = 1). (**D**) The percentage of *Tgfb1* gene expression in different clusters (n = 1). (**E**) TGFβ1 protein expression in contralateral brain with MCAO at RP2D (n=3,20 slices/mouse). (**F**) TGFβ1 protein expression in the ipsilateral brain with MCAO at RP2D, RP4D, and RP8D (n = 3, 20 slices/mouse). (**G**) Immunofluorescence staining of CD45 and *Tgfb1* gene expression in WT mice injected with anti-Ly6C/Ly6G after MCAO at RP2D and quantitative analysis of the proportion of CD45$^+$&*Tgfb1*$^+$ cells in the unit area (n = 4,10 slices/mouse). (**H**) Immunofluorescence staining of CD45 and TGFβ1 protein expression in WT mice injected with anti-Ly6C/Ly6G after MCAO at RP2D. Data are presented as mean ± SEM, unpaired two-tailed Student's t-test (A and G).

The online version of this article includes the following figure supplement(s) for figure 7:

**Figure supplement 1.** Myeloid cells are leading immune cells in the mouse brain after stroke at RP2D.

---

dextran-rhodamine B (~70 kDa) (*Figure 9—figure supplement 1B*) and Texas Red (~71 kDa) leakage was detected (*Figure 9—figure supplement 1C*). The elimination of E-pericytes allowed Evans blue and trypan blue to cross the blood vessel, which indicated that E-pericytes maintain BBB integrity.

Immunofluorescence statistics revealed a reduction in the number of NeuN$^+$ neurons in mice with EC-specific loss of the *Tgfbr2* gene (*Figure 9E*), which led to much more severe neurological deficits. Specifically, EC-specific loss of the *Tgfbr2* gene in mice decreased the latency to fall from the rotarod (*Figure 9F*), increased the frequency of falling from the pole (*Figure 9G*), increased the deviation from normal in the corner test (*Figure 9H*), and impaired the removal of adhesive tape from the forepaw (*Figure 9I*). The TGFβ1-TGFβR2 pathway induces the transdifferentiation of ECs into pericytes, contributing to BBB restoration and neuron protection after stroke.

## EC-specific overexpression of the *Tgfbr2* gene decreases BBB leakage and facilitates neurological recovery after stroke

The elimination of E-pericytes by DTA exacerbated BBB leakage, decreased CBF, and impeded neurological and spontaneous behavioral recovery. Thus, increasing the number of E-pericytes might accelerate BBB repair, increase CBF, alleviate neurological deficits, and promote spontaneous behavioral recovery after stroke. EC-specific overexpression of the *Tgfbr2* gene via AAV2/9-BI30-EF1α-DIO-Tgfbr2-3XFLAG-P2A-DsRed-WPREs was found to promote E-pericyte generation (*Figure 6G and H*). Additionally, EC-specific overexpression of the *Tgfbr2* gene significantly increased CBF recovery (*Figure 10—figure supplement 1A and B*) and effective perfusion in the ischemic brain region at RP34D (*Figure 10—figure supplement 1C and D*). Thus, EC-specific overexpression of the *Tgfbr2* gene promotes CBF restoration after stroke, through the generation of E-pericytes.

EC-specific overexpression of the *Tgfbr2* gene by a virus (Tgfbr2) increased the number of ECs and pericytes (*Figure 10—figure supplement 1E*) and renewed the EC/pericyte ratio close to normal (*Figure 10—figure supplement 1E*). EC-specific overexpression of the *Tgfbr2* gene also decreased Evans blue leakage into the ischemic area, although no obvious Evans blue signal was detected under bright-field microscopy (*Figure 10A*). Moreover, brain atrophy was reduced in the EC-specific overexpression of the *Tgfbr2* gene group (*Figure 10A*).

Immunofluorescence statistics revealed an increase of NeuN$^+$ neurons after stroke in EC-specific overexpression of the *Tgfbr2* gene group, indicating that more neurons survived (*Figure 10B*). Neurological deficits were assessed by evaluating locomotion at different reperfusion times. EC-specific overexpression of the *Tgfbr2* gene increased the latency to fall from the rotarod (*Figure 10C*) and decreased the frequency of falling from the pole (*Figure 10D*). Furthermore, EC-specific overexpression of the *Tgfbr2* gene decreased the deviation from normal in the corner test (*Figure 10E*) and improved the ability of mice to remove adhesive tape from the forepaws (*Figure 10F*). Increasing the number of E-pericytes by EC-specific overexpression of the *Tgfbr2* gene reduces BBB leakage, increases CBF, promotes angiogenesis, protects neurons, and enhances brain self-recovery after stroke.

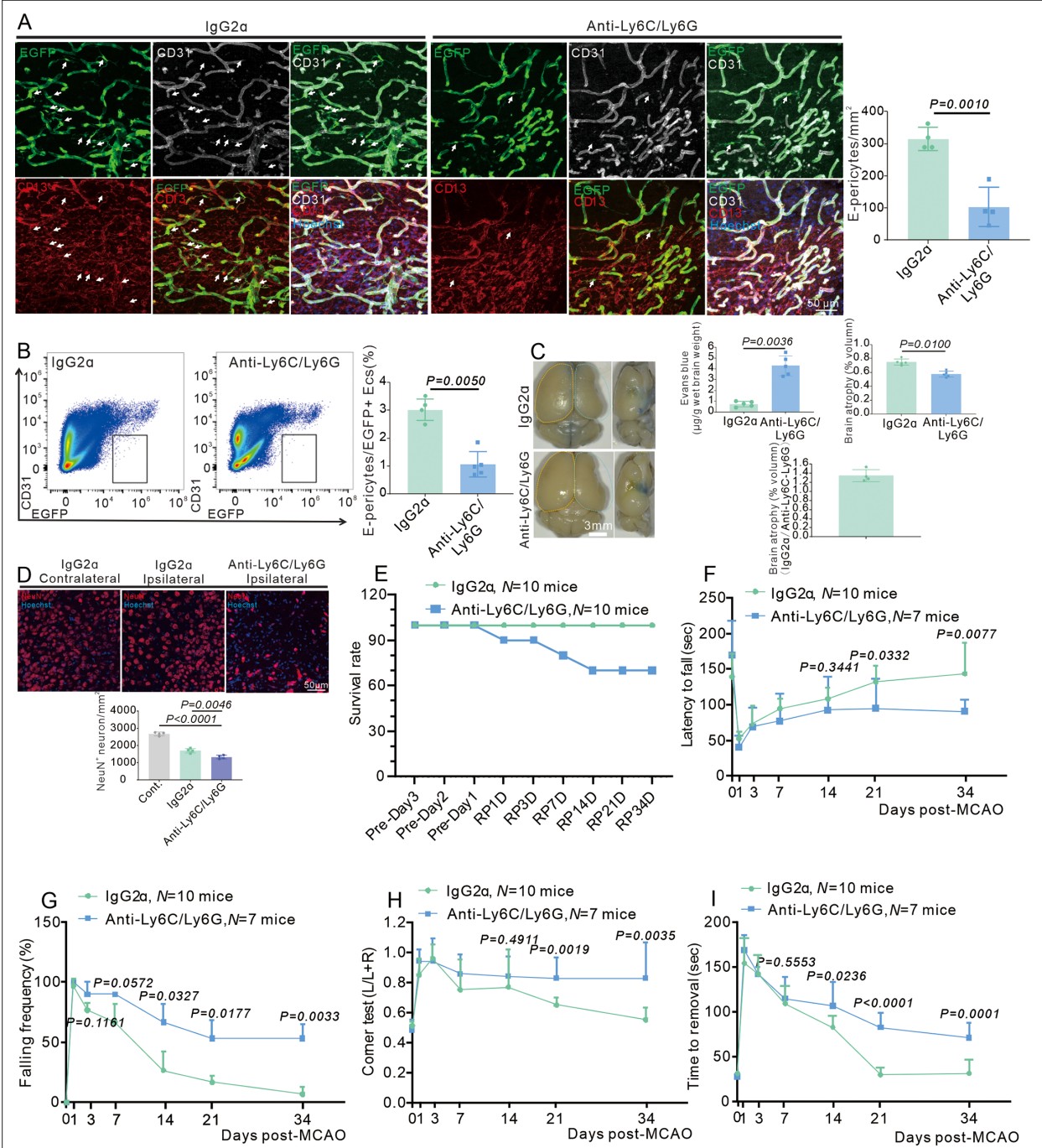

**Figure 8.** Infiltrating myeloid cells promote blood-brain barrier (BBB) renovation and neurological recovery after stroke. (**A**) Immunofluorescence staining of CD13 and CD31 expression in wild-type (WT) mice injected with anti-Ly6C/Ly6G at RP34D after middle cerebral artery occlusion (MCAO) and quantitative analysis of the number in the unit area (n = 4, 20 slices/mouse). (**B**) Flow cytometry analysis of the proportion of EGFP+&CD31+ cells in WT mice injected with anti-Ly6C/Ly6G at RP34D after MCAO and quantitative analysis of the percentage (n = 4). (**C**) Image showing the leakage of Evans blue in WT mice injected with anti-Ly6C/Ly6G at RP34D after MCAO and quantitative analysis of the leakage of Evans blue and brain atrophy volume (n = 5). (**D**) Immunofluorescence staining of NeuN expression in WT mice injected with anti-Ly6C/Ly6G at RP34D after MCAO and quantitative analysis of the number in the unit area (n = 4, 20 slices/mouse). (**E**) Graph showing the survival rate of mice in WT mice injected with anti-Ly6C/Ly6G at RP34D after MCAO (n = 10). (**F**) Graph showing rotarod test in WT mice injected with anti-Ly6C/Ly6G at RP34D after MCAO (n = 7–10). (**G**) Graph showing beam walking test in WT mice injected with anti-Ly6C/Ly6G at RP34D after MCAO (n = 7–10). (H) Graph showing corner test in WT mice injected with anti-Ly6C/Ly6G at RP34D after MCAO (n = 7–10). (**I**) Graph showing adhesive movement test in WT mice injected with anti-Ly6C/Ly6G at RP34D after MCAO (n = 7–10). Data are presented as mean ± SEM, unpaired two-tailed Student's t-test (A-D and F-I).

*Figure 8 continued on next page*

*Figure 8 continued*

The online version of this article includes the following figure supplement(s) for figure 8:

**Figure supplement 1.** Infiltrating myeloid cells promote vascular function recovery after stroke.

## Discussion

After stroke, EC-pericyte transdifferentiation promotes BBB recovery, CBF increase, and neurological recovery, with infiltrating myeloid cells driving E-pericyte generation to enhance these restorative processes.

In other tissues, myeloid cells have been reported in tissue repair. Monocytes regulated hypodermal adipocytes and associated leptin-mediated revascularization of wounds post-infection (*Kratofil et al., 2022*). Macrophages actively secreted metabolites during regeneration to establish immune-muscle stem cells (MuSCs) cross talk in a GSDMD-dependent manner to promote tissue repair (*Chi et al., 2024*). Cerebral ischemic injury and repair involve both cerebral cells and the immune system, which play a critical role in determining stroke outcomes. Studies had shown that *CCR2* knockout and pharmacological inhibition of CCR2 attenuated MDM-induced acute brain injury (*Fang et al., 2018*). However, MDMs recruited to the injured brain after ischemic stroke contribute to long-term spontaneous functional recovery during the chronic stage (*Fang et al., 2018*; *Wattananit et al., 2016*). The ability of MDMs to promote functional recovery was because macrophages were polarized toward the anti-inflammatory phenotype and promoted angiogenesis during the recovery stage (*Fang et al., 2018*; *Wattananit et al., 2016*). However, the precise role of myeloid cells in angiogenesis and spontaneous behavioral recovery after stroke remains to be elucidated. Inhibiting the infiltration of myeloid cells into the brain could reduce the number of CD31$^+$ ECs, implying the decrease of angiogenesis. Furthermore, the suppression of myeloid cell-derived TGFβ1 in the brain restrained endothelial-to-pericytic transition and decreased the pool of pericytes. Pericytes play a vital role in angiogenesis and maintaining the stability of newly formed blood vessels. Myeloid cells promote angiogenesis by driving endothelial-to-pericytic transition. Their inhibition reduces pericyte numbers, ultimately impairing angiogenesis and CBF.

During the acute phase of stroke, ischemia and hypoxia lead to rapid death of pericytes, which increases BBB leakage (*Armulik et al., 2010*). Although the decrease of pericytes during the acute phase of stroke has been reported, there is almost no research on the pericyte pool during the chronic phase. During the chronic phase of stroke, the pericyte pool gradually increased, including both the self-proliferation of pericytes and the conversion of ECs to pericytes. E-pericytes were crucial for the function of vascular vessels during brain self-repair. Depletion of E-pericytes led to the loss of vascular self-recovery function, including BBB integrity and CBF recanalization, and impaired neurological recovery. Increasing the number of E-pericytes reduced Evans blue leakage, augmented CBF, and promoted neurological recovery, which suggested the importance of E-pericytes for blood vessel self-recovery.

Unlike the liver and skin, owing to powerful self-healing ability, stroke can lead to severe disability due to the limited capacity of cerebral cells to regenerate or replace dead cells, especially neurons. Exploring and confirming therapeutic theory for neurons and other cells protection is necessary for stroke. Synapse-level reconstruction of neural circuits and remyelination are involved in the recoverable process via various neurotrophic factors and cell transplants, which also are confronted with rigorous challenges to maintain long-term survival because of lack of effective blood flow (*Svara et al., 2022*; *Sanes and Zipursky, 2020*). A novel pathway for brain self-repair during the chronic phase of stroke was identified. ECs can give rise to pericytes to replenish the pool, accelerating the restoration of BBB function. E-pericytes stimulated angiogenesis and reversed the no-reflow phenomenon, leading to restored blood flow and elevated CBF in the ischemic brain. All the above improvements are beneficial in promoting neuron survival and functional recovery after stroke.

There is no doubt that the brain also has a limited ability to repair itself, which is not as strong as the liver and skin, but if the brain's ability to repair itself can be amplified, it will be huge progress for the therapy of stroke. The role of TGFβ family members in angiogenesis is underscored by the observations that nearly all mice with knockout of specific TGFβ family members died during midgestation due to yolk sac angiogenesis defects (*Goumans et al., 2009*). Mutations in the genes encoding TGFβ pathway components are linked to an increasing number of human pathologies involving vascular

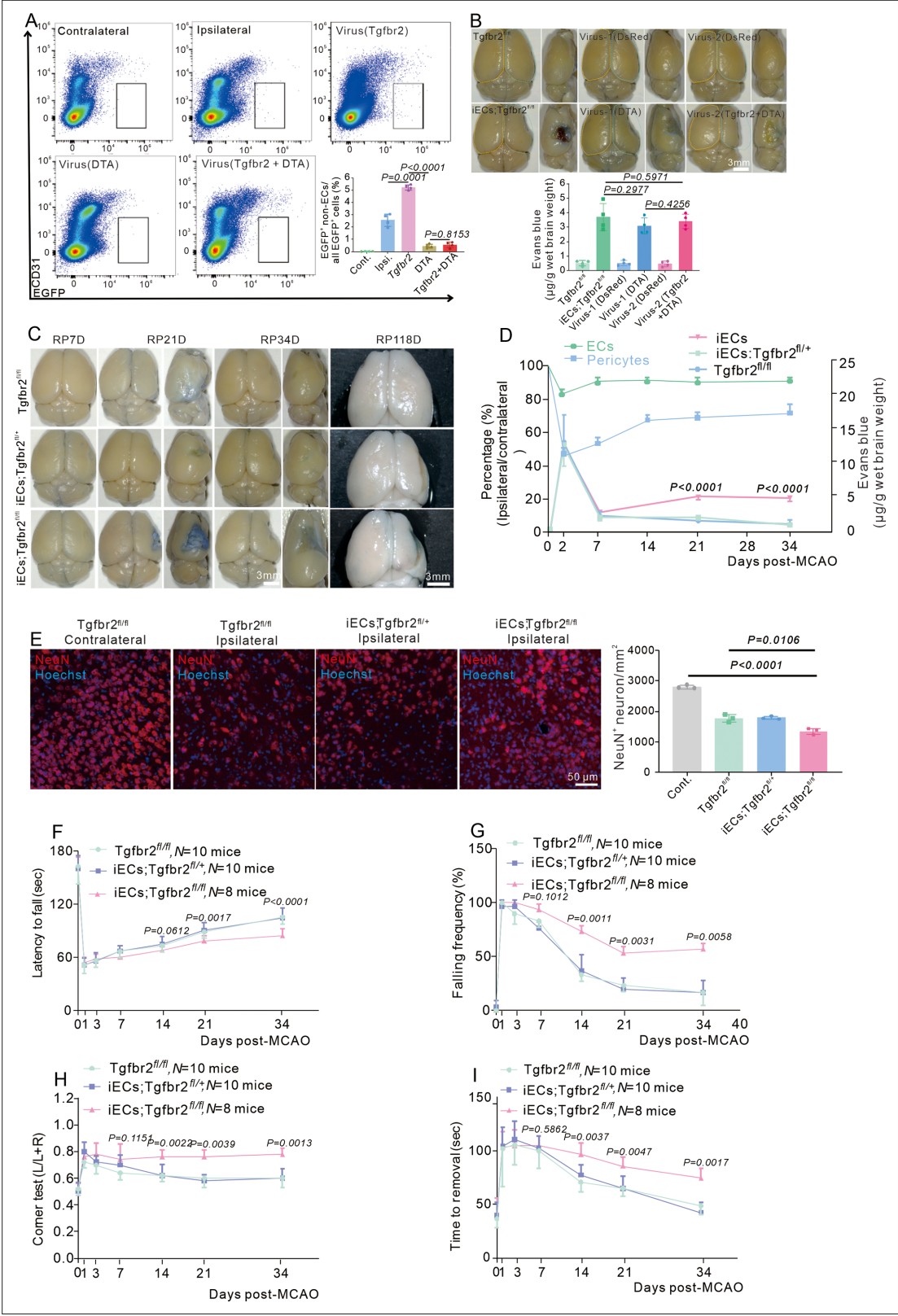

**Figure 9.** E-pericytes deletion by specific endothelial cells (ECs) knockout of the *Tgfbr2* gene aggravates blood-brain barrier (BBB) leakage and neurological deficit after stroke. (**A**) Flow cytometry analysis of the proportion of E-pericytes synchronous injection with AAV2/9-BI30-DIO-NG2-promotor diphtheria toxin A (DTA) virus and AAV2/9-BI30-EF1α-DIO-Tgfbr2-3XFLAG-P2A-DsRed-WPREs virus after middle cerebral artery occlusion (MCAO) at 14D (n = 4). (**B**) Image showing the leakage of Evans blue in induced ECs (iECs);Tgfbr2fl/fl mice at different times after MCAO and quantitative analysis

*Figure 9 continued on next page*

*Figure 9 continued*

of the leakage of Evans blue (n = 3). (**C**) Graph showing the leakage of Evans blue in iECs;Tgfbr2$^{fl/fl}$ mice at different times after MCAO and quantitative analysis of the leakage of trypan blue (n = 3). (**D**) Graph showing the percentage of pericyte and ECs, the leakage of Evans blue in iECs;Tgfbr2$^{fl/fl}$ mice at different times after MCAO (n = 3–6). (**E**) Immunofluorescence staining of NeuN expression in iECs;Tgfbr2$^{fl/fl}$ mice at RP34D after MCAO and quantitative analysis of the number of neurons in the unit area (n = 3, 20 slices/mouse). (**F**) Graph showing the rotarod test in iECs;Tgfbr2$^{fl/fl}$ mice after MCAO at RP34D (n = 8–10). (**G**) Graph showing the beam walking test in iECs;Tgfbr2$^{fl/fl}$ mice after MCAO at RP34D (n = 8–10). (**H**) Graph showing the corner test in iECs;Tgfbr2$^{fl/fl}$ mice after MCAO at RP34D (n = 8–10). (**I**) Graph showing the adhesive movement test in iECs;Tgfbr2$^{fl/fl}$ mice after MCAO at RP34D (n = 8–10). Data are presented as mean ± SEM, unpaired two-tailed Student's t-test (A-B and D-I).

The online version of this article includes the following figure supplement(s) for figure 9:

**Figure supplement 1.** E-pericytes deletion by specific endothelial cells (ECs) knockout of the *Tgfbr2* gene aggravates blood-brain barrier (BBB) leakage of small molecules.

dysfunction (*ten Dijke and Arthur, 2007*). Increasing TGFβR2 expression could increase the number of E-pericytes, reduce BBB leakage, increase CBF, and accelerate neurological functional recovery. Thus, the TGFβ1-TGFβR2 pathway is a potential therapeutic agent for promoting brain self-repair after stroke. At the same time, ECs can be transformed into pericytes in other tissues after ischemia, suggesting that activation of the TGFβ1-TGFβR2 pathway may also be involved in the repair of other tissues.

In summary, transdifferentiation from ECs to pericytes is identified post-stroke, mediating both cellular reprogramming and recovery of BBB integrity and neurological function. E-pericytes are driven by infiltrating myeloid cells, which release TGFβ1 to induce ECs subjected to EndoMT. The discovery of E-pericytes in fueling the destructive pericyte pool to remedy the functions of the lost pericytes after stroke indicates the capacity of brain self-repair from cell transformation.

## Limitations of the study

After stroke, ECs can be converted into E-pericytes. The TGFβ1-TGFβR2 pathway has been shown to induce EndoMT in ECs, leading to the formation of fibroblast-like cells at RP8D. The phenomenon has been confirmed in other disease models. However, the conversion of fibroblast-like cells into pericytes has been rarely reported. The mechanisms by which fibroblast-like cells convert into pericytes remain unexplored, including the key transcription factors involved and whether their manipulation could specifically regulate E-pericytes to promote post-stroke recovery. Some E-pericytes are found to migrate from blood vessels, while others adhere to blood vessels. Based on the normal positioning of pericytes, E-pericytes located on blood vessels are expected to function more effectively. Therefore, how to promote the return of migrating E-pericytes to blood vessels remains an unsolved problem.

## Methods

### Experimental design

Transgenic Cdh5CreERT2 transgenic mice cross with reporter mice *Ai47 (EGFP $^{fl/fl}$) reporter* or *Ai14 (tdTomato$^{fl/fl}$) reporter* mice. Thus, ECs are labeled in green or red initially. TAM was delivered when mice were 1.5 months of age (6 weeks), i.e., genetic tracing started. After waiting for 2 weeks, MCAO was performed. After a 2 hr occlusion, reperfusion began. The contralateral sides of the ischemic brain were used as objects for homeostasis conditions. Thus, it was traced for 22 days in PR8D mice (N=3 mice), 1.6 months in RP34D (N=27 mice), 2.8 months in RP71D (N=3 mice), 8.2 months in RP232D (N=3 mice), 14 months in RP408D (N=2 mice), 17.6 months in RP51D mice (N=2 mice).

### Animal care and tissue dissection

All animal experiments were carried out following protocols approved by the Institutional Animal Care and Use Committee (IACUC) at the School of Life Sciences, Westlake University. Wild-type C57BL6/J mice were purchased from the Laboratory Animal Resources Center of Westlake University. Cdh5CreERT2 mice were a gift from Le-Ming Zheng (Peking University). Gt (ROSA)26Sor$^{tm47(CAG-EGFP*)Hze}$ (Ai47) was shared by Zilong Qiu (ION). Tie2Dre and Mfsd2aCrexER mice were a gift from Bin Zhou (Chinese Academy of Sciences). Tgfbr2$^{fl/fl}$ mice were a gift from Bing Zhang (Westlake University). We crossed Cdh5Cre-ERT2 mice to Ai47 mice to generate Cdh5CreERT2:Ai47 mice. Standard chow and water were provided to mice ad libitum. Four mice were housed in each cage. All animals were

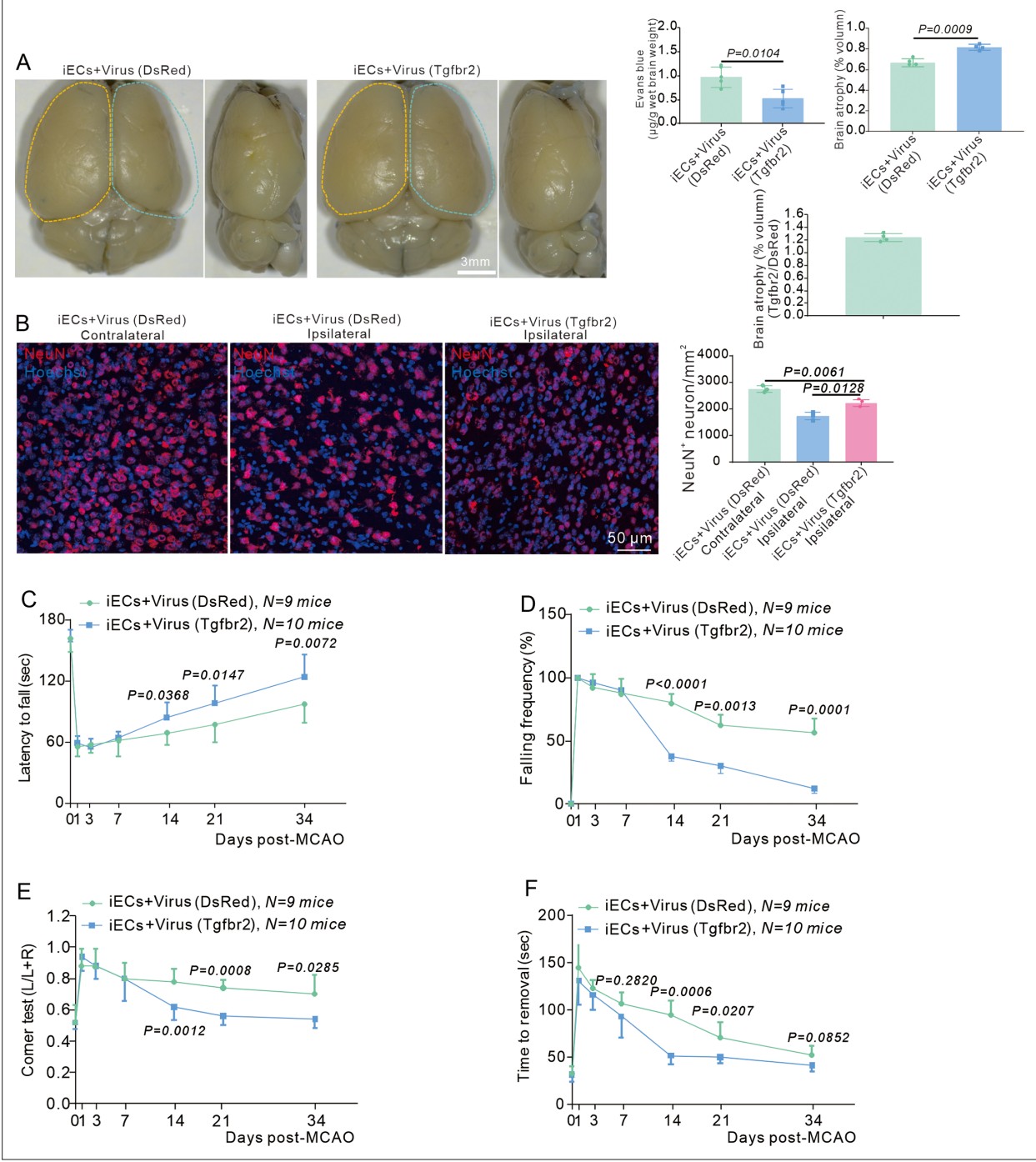

**Figure 10.** Endothelial cell (EC)-specific overexpression of the *Tgfbr2* gene reinforces blood-brain barrier (BBB) function and neurological recovery after stroke. (**A**) Image showing the leakage of Evans blue in Cdh5CreERT2 injected with AAV2/9-BI30-EF1α-DIO-Tgfbr2-3XFLAG-P2A-DsRed-WPREs virus at RP34D after middle cerebral artery occlusion (MCAO) and quantitative analysis of the leakage of Evans blue (n = 5). (**B**) Immunofluorescence staining of NeuN expression in Cdh5CreERT2 mice injected with AAV2/9-BI30-EF1α-DIO-Tgfbr2-3XFLAG-P2A-DsRed-WPREs virus after MCAO at RP34D and quantitative analysis of the number in the unit area (n = 3, 20 slices/mouse). (**C**) Graph showing rotarod test in Cdh5CreERT2 mice injected with AAV2/9-BI30-EF1α-DIO-Tgfbr2-3XFLAG-P2A-DsRed-WPREs virus after MCAO (n = 8–10). (**D**) Graph showing beam walking test in Cdh5CreERT2 mice injected with AAV2/9-BI30-EF1α-DIO-Tgfbr2-3XFLAG-P2A-DsRedWPREs virus after MCAO (n = 8–10). (**E**) Graph showing corner test in Cdh5CreERT2 mice injected with AAV2/9-BI30-EF1α-DIO-Tgfbr2-3XFLAG-P2A-DsRed-WPREs virus after MCAO (n = 8–10). (**F**) Graph showing adhesive movement test in Cdh5CreERT2 mice injected with AAV2/9-BI30-EF1α-DIO-Tgfbr2-3XFLAG-P2A-DsRed-WPREs virus after MCAO (n = 8–10). Data are presented as mean ± SEM, unpaired two-tailed Student's t-test (A-F).

The online version of this article includes the following figure supplement(s) for figure 10:

**Figure supplement 1.** Endothelial cell (EC)-specific overexpression of the *Tgfbr2* gene reinforces cerebral blood flow (CBF) and vessel length.

housed in a standard animal room with a 12/12 hr light/dark cycle at 25°C. Both male and female mice were used in the study. Mice were anesthetized by intraperitoneal injection of pentobarbital sodium (100 mg/kg), and brain dissection was performed immediately for FACS experiments, or cardiac paraformaldehyde (PFA) perfusion was performed for the immunohistochemistry experiment. After fixation, brains were dissected and kept in 4% PFA for an additional 4 hr and then transferred to PBS until further processing.

## MCAO

Adult mice were anesthetized with pentobarbital sodium (100 mg/kg), and body temperature was maintained during surgery with a heating pad. A midline neck incision was made, the right common carotid artery was carefully separated from the vagus nerve, and the artery was ligated using a 5.0 string. A second knot was made on the left external carotid artery. The right internal carotid artery (ICA) was isolated, and a knot was left loose with a 5.0 string. The knot was not tightened until the intraluminal insertion was done. A small hole was cut in the common carotid artery before it was bifurcated to the external carotid artery and ICA. A silicon-coated monofilament (tip diameter = 230 μm, Doccol Corporation, 602256PK5Re) was gently advanced into the ICA until it stopped at the origin of the MCA in the circle of Willis. The third knot on the ICA was closed to fix the filament in position. During MCA occlusion (2 hr), mice were kept in a warm cage at 25°C.

## Laser speckle contrast imaging

Laser speckle contrast imaging provides a measure of CBF by quantifying the extent of blurring of dynamic speckles caused by the motion of red blood cells through the vessels. Briefly, mice were placed under an RFLSI III device (RWD Life Sciences) before and after the suture was successfully inserted through the CCA. The skull over both hemispheres was exposed by making an incision along the midline of the scalp. When a 785 nm laser is used to illuminate the brain, it produces a random interference effect that represents blood flow in the form of a speckle pattern. Scattering light was detected by a charge-coupled device camera, and the images were acquired by custom software from RWD Life Sciences Company. For each acquisition, a total of 160 images, each of which measured 2048×2048 pixels, were collected at 16 Hz.

## Laser Doppler flowmetry

After the laser emitted by the LDF device is irradiated onto the tissue, the laser is scattered by the moving red blood cells in the blood, causing a change in the frequency of the scattered light. The frequency change is related to the speed of the red blood cells, so the blood flow velocity can be inferred by measuring the Doppler shift. Briefly, mice were placed under an LDF device (moor-VMS-LDF1) before and after the suture was successfully inserted through CCA at RP34D. Connect the probe to the Laser Doppler Flowmeter, turn on the device to start measuring, determine the brain area you wish to measure, position the Laser Doppler probe on the desired brain area, record the data, export the raw data and analysis.

## Tamoxifen

Tamoxifen, 10 mg/ml dissolved in corn oil, was administered intragastrically with 200 μl for 4 consecutive days. Before further experiments, fluorescent protein expressions in mouse ear vasculatures were first examined. Tamoxifen was administered at least 2 weeks before MCAO surgery.

## Immunohistochemistry

Cryosections of fixed mouse brains (50 μm) were handled free-floating and washed for 5 min in PBS before further procedures. Tissue sections were permeabilized and blocked in the PBS containing 0.5% Triton and 5% BSA at room temperature (RT) for 1 hr. Then, brain sections were incubated with different primary antibodies, including CD31 (1:400, BD, Cat# 557355, RRID:AB_396660), VE-Cadherin (1:200, BiCellScientific, Cat# 00105, RRID:AB_3076926), ERG (1:300, Sigma, Cat# ab110639, RRID:AB_10861578), GLUT1 (1:300, Sigma, Cat# 07-1401, RRID:AB_11213098), NeuN (1:300, Merck, Cat# ABN90P, RRID:AB_11205760), Doublecortin (1:300, Santa Clus, Cat# sc-271390, RRID:AB_10609969), Iba1 (1:300, HUABIO, Cat# ET1705-78, RRID:AB_2920891), GFAP (1:300, Proteintech, Cat# CL488-60190, RRID:AB_2920892), PDGFRβ (1:300, Thermo Fisher, Cat# 14-1402-82, RRID:AB_11213099,

RRID:AB_11213099), NG2 (1:300, Merck, Cat# AB5320, RRID:AB_11213100), CD68 (1:300, Bio-Rad, Cat# MCA1957T, RRID:AB_322219), CD45-APC (1:200, BioLegend, Cat# 103112, RRID:AB_312977), Ki67 (1:300, Thermo Scientific, Cat# RM-9106-S0, RRID:AB_2341197), F4/80 (1:300, Cell Signaling, Cat# 30325, RRID:AB_2798963), CD13 (1:300, R&D Systems, Cat# AF2335, RRID:AB_354773), α-SMA-CY3 (1:300, Sigma, Cat# C6198, RRID:AB_476856), which were all diluted in the blocking solution (1% percelin, 5% BSA, and 0.1% Triton in PBS) at 4°C overnight. After rinsing in PBS three times for 5 min each, brain slices were incubated with secondary antibodies accordingly, including donkey anti-mouse IgG (A21202, Thermo Fisher Scientific, 1:1000) and donkey anti-rabbit IgG (A10040, Thermo Fisher Scientific, 1:1000). They were diluted in the blocking solution. After a 2 hr incubation, brain slices were washed with PBS three times for 15 min each. Finally, Hoechst counterstaining was performed for each specimen. After mounting in an anti-fade mounting medium, images were acquired by using a Zeiss LSM800 laser-scanning confocal microscope with a ×10, ×20, or ×63 objective, with or without Airyscan mode.

## Flow cytometry

Primary cell preparation was conducted as described below. Mouse brains were collected into cold HBSS with glucose and BSA (HBSS, Vendor info) and then were cut into small pieces by sterile scissors, followed by a centrifuge at 1600 rpm/min for 5 min to get rid of supernatant. The minced tissues after the centrifuge were incubated with collagenase IV (2.5 mg/ml, Roche, Cat# 11088858001) at 37°C with gentle rotation for 30 min in 2 ml of HBSS. Next, samples were passed through a 70 µm filter and washed with HBSS. To obtain higher cell yield, grind the tissue on the 70 µm filter using a syringe piston and guarantee all tissue passes the filter. The following antibodies in 1:200 dilutions were used: CD31-PE-594 (1:200, BioLegend, Cat# 102520, RRID:AB_2563241), CD45-APC (1:200, BioLegend, Cat# 103112, RRID:AB_312977), Ly6C-APC (1:200, BioLegend, Cat# 128016, RRID:AB_1732079), CD11b-PE (1:200, BioLegend, Cat# 101212, RRID:AB_312791), Viability dye (1:1000, Thermo Fisher, Cat# 65-0863-14, RRID:AB_2861190). Antibody incubation was performed on ice for 30 min in HBSS with glucose and BSA. All analyses were performed with CytoFLEX LX-5L1 (Beckman). FlowJo V10 software was used for data analyses.

## EdU injection

EdU (200 mg/kg, Alfa Aesar Chemical) was injected intraperitoneally for 34 days since MCAO was performed. EdU was initially dissolved in dimethylsulfoxide (DMSO) at the stock concentration of 500 mg/ml and was diluted with saline before injection.

## Calculation of brain atrophy volume

Fixed mouse brains were cut with 100 µm and photographed using Axio Zoom.V16. Using ImageJ software, the image is converted to a grayscale image. Click Analyze, then Set Measurements, select Area, and select Show Brain Slice Outline. Calculate the area of each brain slice. Cerebral atrophy volume ratio = (sum of white ischemic area of each section)/(sum of brain area of each section)×100%.

## Calculation of signal length and intensity in immunofluorescence

Fixed mouse brains were cut with 50 µm, stained with relevant antibodies by immunofluorescence, and photographed using Zeiss LSM 800 Confocal Laser Scanning Microscopy (×20). Using ImageJ software, the single signal in the image is converted to a grayscale image and calculated in length and intensity. The percentage of targeted signal length or intensity/CD31$^+$ blood vessel length shows the targeted signal expression in blood vessels.

## Calculation of the number of cells in immunofluorescence

Fixed mouse brains were cut with 50 µm, stained with relevant antibodies by immunofluorescence, and photographed using Zeiss LSM 800 Confocal Laser Scanning Microscopy (20×, 0.1 mm$^2$). After maximum intensity projection, the number of cells was calculated at 0.1 mm$^2$.

## TGFβR2 inhibitors injection

LY-364947 (Sigma, L6293) and SB-431542 (SelleckBio, S1067) were dissolved in DMSO. For the in vivo inhibition of the TGFβ signaling, LY-364947 (10 mg/kg) and SB-431542 (10 mg/kg) were

intraperitoneally injected daily starting MCAO. The control mice were treated in parallel with the vehicle only. The animals were killed at RP34D.

## Single-cell workflow

scRNA-seq was performed using a 10x Next GEM Single-Cell 3'GEM kit v3.1 (10x Genomics) according to the manufacturer's protocol. Briefly, an equal number of FACS-isolated CD45$^-$&Viability$^-$&EGFP$^+$ cells (sorting for ECs) and CD45$^+$& Viability$^-$ cells (sorting for immune cells) from the brain. ~1000 cells per µl were immediately loaded into the 10x Chromium controller. Separate 10x Genomics reactions were used for each group. Generated libraries were sequenced on an Illumina NovaSeq with >27.5 × 10$^3$ reads per cell followed by demultiplexing and mapping to the mouse genome (build mm10) using CellRanger v5.0 (10x Genomics). Gene expression matrices were generated using the CellRanger software v5.0.1 (10x Genomics) with standard settings and mapping to the mm10 reference mouse genome. The following data analysis was performed using the Seurat package v3.0.

## Bulk RNA-seq analysis

Messenger RNAs used for bulk RNA-seq were harvested from the sorted parenchymal EGFP$^+$&CD31$^+$ cells (contralateral and ipsilateral) and EGFP$^+$& CD31$^-$ cells (ipsilateral) from adult Cdh5CreERT2:Ai47 mice, which received tamoxifen as adults. RNA was isolated using the RNeasy Plus Mini Kit (QIAGEN). RNA-seq libraries were prepared using a TruSeq RNA Library Prep kit v2 (Illumina). The libraries were sequenced on a HiSeq 2500 instrument with single-end 300–400 bp reads (single indexing reads). The normalized gene count matrix for all genes was used as input through the htseq-count script. A heatmap was generated using R code to visualize the gene expression levels in different groups. The genes of interest were clustered as neuromediator receptors and ranked according to expression level from high to low. The DESeq2 was used to compare differences in gene expression between different groups.

## *Tgfb1* gene RNAscope

To show which cells synthesize the secreted protein TGFβ1, we used RNAscope to test. Briefly, the brains were fixed in chilled 4% PFA for 8 hr at 4°C, washed twice in 1× PBS, and dehydrated with 15% sucrose and 30% sucrose overnight. Mice's brains were cut with 14 µm. The following day, sections were washed twice in 50 µl of 2× saline sodium citrate buffer (SSC) for 10 min. smFISH Probes (Advanced Cell Diagnostics) were preheated for 10 min at 40°C, cooled to RT, and added to sections to incubate (always in a humidified chamber) for 2 hr at 40°C. Probes were used to target *Tgfb1* mRNA (ACD Cat No. 406201, NM_009370.2). Following probe incubation, sections were washed (always with RNAscope wash buffer for 1 min) four times, incubated in RNAscope AMP-1 solution for 30 min at 40°C, washed four times, incubated in RNAscope AMP-2 solution for 15 min at 40°C, washed four times, incubated in RNAscope AMP-3 solution for 30 min at 40°C, washed four times, incubated in RNAscope AMP-4 solution for 15 min at 40°C and washed four times. For concomitant immunostaining, sections were incubated in 5% BSA blocking buffer for 1 hr at RT, washed once with 1× PBS, and incubated in primary antibody solution (CD31 [1:400, BD, Cat# 557355, RRID:AB_396660], Iba1 [1:300, HUABIO, Cat# ET1705-78, RRID:AB_2920891], CD13 [1:300, R&D Systems, Cat# AF2335, RRID:AB_354773]), overnight at 4°C. The following day, sections were washed three times with 1× PBS for 5 min each and incubated in fluorescently labeled secondary antibody solution diluted 1:1000 in 1× PBS in the dark for 2 hr at RT. Sections were washed three times in 1× PBS for 5 min, incubated in 0.5 mg/ml Hoechst 33258 (Sigma-Aldrich) for 2 min at RT, washed once with 1× PBS for 5 min, and mounted on glass slides using PermaFluor (Thermo Fisher). Sections were imaged on a confocal microscope (Zeiss LSM 800).

## Deletion of myeloid cells

For myeloid cells depletion, mice received intraperitoneal injections of 400 µg of anti-Ly6C (InVivoMAb Antibodies, BE0203) and anti-Ly6G (InVivoMAb Antibodies, BP0075) antibodies per mouse starting 2 days ahead before MCAO, then received intraperitoneal injections of 100 µg at RP0D, RP2D, RP4D, RP6D, and RP8D. Myeloid cells were monitored by FACS at RP2D and RP8D to evaluate the functions of anti-Ly6C and anti-Ly6G.

## Measurement of BBB integrity

To evaluate the leakage of Evans blue and trypan blue, the mice were injected with 100 µl of 2.5% Evans blue (Sigma-Aldrich, St. Louis, MO, USA) and 0.25% trypan blue (Sigma-Aldrich, 93595) intravenously through the tail vein 1 hr before fixing using 4% PFA. Then, the brains were removed and imaged under microscopy. The brain was then separated into ipsilateral and contralateral hemispheres. Next, each hemisphere was supplemented with 500 µl of trichloroacetic acid (TCA), transferred to a 55°C heat block, and incubated for 24 hr to extract Evans blue from the tissues. The mixture was centrifuged to pellet any remaining tissue fragments, and absorbance was measured at 610 nm; 500 µl TCA was used as a blank. The Evans blue extravasated per g tissue was determined. 71KD-Texas Red (10 mg/ml, Vector Laboratories, TL-1176) and 70KD-Rhodamine B isothiocyanate-Dextran (10 mg/ml, Sigma, R9379) were also injected with 100 µl by tail vein injection, and mice brain fixed with 4% PFA after 1 hr later.

## Rotarod test

Rotarod test protocol execution by double-blind is used to assess motor coordination and balance in mice. Place the mouse on a rotating rod. The rod gradually accelerates from a low speed to a higher speed (from 4 to 40 rpm in 300 s). The mouse must maintain balance and motor coordination to stay on the rod. Record the time (latency) it takes for the mouse to fall off the rod. Repeat the test three times to get consistent results. Before the test, train the mice for 3 days, and the goal is for the mice to be able to walk forward on the rotating rod. Rotarod test recorded on post-stroke 1, 3, 7, 14, 21, and 34 days for three times every day with 1 hr intervals.

## Rotating beam test

The rotating beam test (RBT) execution by double-blind is used to evaluate the motor, balance, and sensory functions of animals. The RBT was performed 3D before the induction of stroke and on post-stroke 1, 3, 7, 14, 21, and 34 days for three times every day with 1 hr intervals. Briefly, before testing, the mice were subjected to training sessions for 3 consecutive days. Each training day consisted of three consecutive sessions where the mice traveled across the entire beam. During the test, the mice were placed on a beam rotating at 3 r.p.m., and the fall frequency, average speed, and total travel distance were recorded and analyzed.

## Corner test

The corner test execution by double-blind is used to detect unilateral abnormalities of sensory and motor functions in the stroke model. Mouse is placed between two boards, each with a dimension of 30×20×1 cm$^3$. The edges of the two boards are attached at a 30° angle with a small opening along the joint between the two boards to encourage entry into the corner. The mouse is placed between the two angled boards facing the corner and halfway to the corner. The nonischemic mouse turns either left or right, but the ischemic mouse preferentially turns toward the non-impaired, ipsilateral (right) side. The turns in one versus the other direction are recorded from 10 trials for each test. A total of 10 trials are recorded per animal preoperatively and on indicated days. Corner test recorded on post-stroke 1, 3, 7, 14, 21, and 34 days for three times every day at 1 hr intervals.

## Adhesive removal test

The adhesive removal test execution by double-blind is widely used in rodents to evaluate sensorimotor dysfunction and motor asymmetry. Briefly, animals are placed into a 15 × 25 cm$^2$ transparent box, and two similar adhesive tapes are attached to the hairless part of each forepaw with the same pressure. The time it takes to contact and remove the stimuli is recorded. In general, animals spend more time contacting and removing the adhesive tape from the contralateral forepaw, while they have no problem contacting and removing the adhesive tape from the ipsilateral forepaw. The adhesive removal test was recorded on post-stroke 1, 3, 7, 14, 21, and 34 days for three times every day at 1 hr intervals.

## Virus injection

The following adenovirus vector AAV2/9-CAG-DIO-EGFP was purchased for ShuMi, Wuhan, China (PT-0168). Half of the 1 µm virus was delivered to postnatal day 2 mouse pups by intracerebroventricular

injection to both sides. After 1.5 months, tamoxifen was administered. One month later, these mice were subjected to MCAO to induce ischemic stroke. The following adenovirus vector AAV2/9-NG2-full length-promoter-DIO-DsRed-WPRES and AAV2/9-NG2-full length-promoter-DIO-DTA-WPRES was purchased for ShuMi, Wuhan, China (PT-9649 and PT-9648). The following adenovirus vector AAV2/9-BI30-EF1α-DIO-Tgfbr2-3XFLAG-P2A-DsRed-WPREs{Virus(Tgfbr2)} was purchased for ShuMi, Wuhan, China (PT-4190). All virus injections were carried via retro-orbital injection with a dosage of $1e*10^{11}$ vg.

## Vascular flow function

At RP34D after MCAO, mice were disposed with cardiac perfusion 9 ml/min with 20 ml of warm (34–37°C) PBS with heparin (20 IU/ml), followed by 20 ml of warm (34–37°C) 0.25% (wt/vol) FITC-Dextran (Sigma-Aldrich, SLCC4853) in 5% (wt/vol) gelatin from porcine skin (Sigma-Aldrich, G1890) in PBS. After placing the mice heads down into ice for 30 min, the brains were extracted and drop-fixed in 4% PFA overnight for immunofluorescence.

## Mouse cranial imaging by two-photon microscopy

Adult mice were anesthetized with pentobarbital sodium, and analgesia was provided by subcutaneous injection of 0.2% meloxicam. The scalp was removed, and the skull was exposed and cleaned. A dental drill with a 0.6-mm-diameter bit was used to engrave and thin the bone around the circular craniotomy area at a size of 3 mm. The piece of skull was carefully peeled off with fine-pointed forceps, and a cranial window was generated. A coverslip with a 3 mm diameter was placed on the cranial window, and its perimeter was completely sealed with a 1% agarose gel. The metal head plate was glued onto the skull with dental acrylic, through which the mouse brain was fixed on a head plate holder. The headplate holder, together with the mouse, was placed under a two-photon microscope (FVMPE-RS, Olympus). The cerebral vasculature *Cdh5CreERT2:Ai47* mice were visualized and imaged once a day before and after stroke. All surgeries were performed with sterilized instruments and an environment.

## Statistics

The quantified data in all figures were analyzed with GraphPad Prism 10.0 (La Jolla, CA, USA) and presented as the mean ± SEM with individual data points shown. Unpaired two-tailed Student's t-test was used to assess the statistical significance between the two groups. Statistical significance was determined by calculation of p-value (*$p<0.05$, **$p<0.01$, ***$p<0.001$, and ****$p<0.0001$, ns: not significant). The repetition of data is independent of biological replicates, and the number of replicates for each experiment is noted in the corresponding figure legend.

## Acknowledgements

We thank L He, W Pei, Z Gao, H Shi, Y Lu, B Cai, and Q Ma for insightful discussions. We thank Y Wang for performing part FACS and bulk RNA-seq library preparation together. We thank D Lu for her responsive and timely purchasing support. We thank Q Gao for part FACS and mouse genotyping together. We thank Y Jin for part Stroke Induced with Magnetic Particles (SIMPLE). We thank P Zhu for a headgear model for the two-photon microscope. We thank J Xie and L Gao for their Virus (AAV-CAG-DIO-EGFP). We thank the animal facility for its technical assistance with rodent housing, and the Biomedical Research Center platform for technique support. JMJ acknowledges the support from the Key R&D Program of Zhejiang (grant 2024SSYS0031), Zhejiang Province Natural Science Foundation (Project # 2022XHSJJ004), the National Natural Science Foundation of China (Projects # 32170961), HRHI programs 202309002 and 202109013 of Westlake Laboratory of Life Sciences and Biomedicine, Westlake University startup funding, the Westlake Education Foundation. The work is also supported by the National Natural Science Foundation of China (Projects # 82101475) to ZZ.

## Additional information

### Funding

| Funder | Grant reference number | Author |
| --- | --- | --- |
| The National Natural Science Foundation of China | Projects # 32170961 | Jie-Min Jia |
| Key R&D Program of Zhejiang | 2024SSYS0031 | Jie-Min Jia |
| Westlake Laboratory of Life Sciences and Biomedicine HRHI program | 202309002 | Jie-Min Jia |
| Westlake Laboratory of Life Sciences and Biomedicine HRHI program | 202109013 | Jie-Min Jia |
| National Natural Science Foundation of China | 82101475 | Zhu Zhu |

The funders had no role in study design, data collection and interpretation, or the decision to submit the work for publication.

### Author contributions

Tingbo Li, Writing – original draft, Writing – review and editing; Ling Yang, Jiaqi Tu, Yufan Hao, Zhu Zhu, Yingjie Xiong, Qingzhu Gao, Lili Zhou, Guanglei Xie, Yiyi Zhang, Bingrui Zhao, Data curation; Dongdong Zhang, Xuzhao Li, Yuxiao Jin, Xi Wang, Conceptualization; Nan Li, Software; Jie-Min Jia, Supervision, Funding acquisition

### Author ORCIDs

Tingbo Li https://orcid.org/0009-0008-2995-0452
Ling Yang http://orcid.org/0009-0009-4763-061X
Jiaqi Tu http://orcid.org/0000-0002-1446-3941
Yufan Hao http://orcid.org/0009-0003-1620-7663
Zhu Zhu http://orcid.org/0000-0001-8552-8469
Yingjie Xiong http://orcid.org/0009-0000-7723-8904
Qingzhu Gao http://orcid.org/0009-0002-1930-9756
Lili Zhou http://orcid.org/0009-0003-4038-2410
Guanglei Xie http://orcid.org/0000-0002-5911-004X
Dongdong Zhang http://orcid.org/0000-0001-5301-4914
Xuzhao Li http://orcid.org/0000-0002-7445-9047
Yuxiao Jin http://orcid.org/0000-0001-9527-8029
Yiyi Zhang http://orcid.org/0009-0009-4436-6307
Bingrui Zhao http://orcid.org/0000-0003-4304-7290
Nan Li http://orcid.org/0000-0001-7826-4688
Xi Wang https://orcid.org/0000-0002-7936-9366
Jie-Min Jia https://orcid.org/0000-0001-8446-3819

### Ethics

All animal experiments were carried out following protocols approved by the Institutional Animal Care and Use Committee (IACUC) at the School of Life Sciences, Westlake University. (24-013-JJM).

Reviewer #1 (Public review): https://doi.org/10.7554/eLife.105593.3.sa1
Reviewer #2 (Public review): https://doi.org/10.7554/eLife.105593.3.sa2
Reviewer #3 (Public review): https://doi.org/10.7554/eLife.105593.3.sa3
Author response https://doi.org/10.7554/eLife.105593.3.sa4

## Data availability

The raw sequencing data generated in the study have been deposited in the NCBI Sequence Read Archive (SRA) under BioProject PRJNA1262095. Plasmids generated in the study will be available through Addgene (Addgene ID 236760, Addgene ID 236761,Addgene ID 236762).

The following dataset was generated:

| Author(s) | Year | Dataset title | Dataset URL | Database and Identifier |
|---|---|---|---|---|
| Westlake University | 2025 | The myeloid cell-driven transdifferentiation of endothelial cells into pericytes promotes the restoration of BBB function and brain self-repair after stroke | https://www.ncbi.nlm.nih.gov/bioproject/?term=PRJNA1262095 | NCBI BioProject, PRJNA1262095 |

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
