## [Editor Report · eLife Assessment]

This **important** study aims to understand the role of endothelial cell differentiation into pericytes in the restoration of blood-brain barrier function after ischemic stroke. Identification of pericytes derived from endothelial cells and the involvement of myeloid cell-derived TGFβ1 signaling are **compelling** new findings, but future studies will be needed to validate the origin and nature of these pericytes. The work will be of interest to blood-brain barrier and basic and translational stroke researchers.

---

## [Referee Report · Reviewer #1 (Public review)]

Summary:

Using lineage tracing and single-cell RNA sequencing, Li et al. reported brain ECs can differentiate into pericytes after stroke. This finding is novel and important to the field.

Strengths:

Detailed characterization of each time point and genetic manipulation of genes for study role of ECs and E-pericyte.

Weaknesses:

Genetic evidence for lineage tracing of ECs and E-pericytes requires more convincing data that include staining, FACS, and scRNA-seq analysis.

Comments on revisions:

Authors have addressed some of my concerns and questions, and also plan to include more convincing data to support the conclusion. Some unpublished data should be included in the online supporting files.

---

## [Referee Report · Reviewer #2 (Public review)]

Summary:

In this manuscript, Li and colleagues study the fate of endothelial cells in a mouse model of ischemic stroke. Using genetic lineage tracing approaches, they find that endothelial cells give rise to non-endothelial cells, which they term "E-pericytes." They further show that depleting these cells exacerbates blood-brain barrier leakage and worsens functional recovery. The authors also provide evidence that endothelial-to-mesenchymal transition, myeloid cell-derived TGFβ1, and endothelial TGFβRII are involved in this process. These are potentially interesting findings, however, the experimental evidence that endothelial cells undergo transdifferentiation to non-endothelial cells is weak, as is the evidence that these cells are pericytes. Addressing this foundational weakness will facilitate interpretation of the other findings.

In this revised manuscript, the authors corrected labeling errors and included negative controls for flow cytometry and immunohistochemistry data. They did not, however, substantively address the major weaknesses below related to rigorously demonstrating the cellular origin and identity of "E-pericytes."

Strengths:

(1) The authors address an important question about blood vessel function and plasticity in the context of stroke.

(2) The authors use a variety of genetic approaches to understand cell fate in the context of stroke. Particularly commendable is the use of several complementary lineage tracing strategies, including an intersectional strategy requiring both endothelial Cre activity and subsequent mural cell NG2 promoter activity.

(3) The authors address upstream cellular and molecular mechanisms, including roles for myeloid-derived TGFβ.

Weaknesses:

(1) The authors use Cdh5-CreERT2; Ai47 mice to permanently label endothelial cells and their progeny with eGFP. They then isolate eGFP+ cells from control and MCAO RP7D and RP34D brains, and use single cell RNA-seq to identify the resulting cell types. Theoretically, all eGFP+ cells should be endothelial cells or their progeny. This is a very powerful and well-conceived experiment. The authors use the presence of a pericyte cluster as evidence that endothelial to pericyte transdifferentiation occurs. However, pericytes are also present in the scRNA-seq data from sham mice, as are several other cell types such as fibroblasts and microglia. This suggests that pericytes and these other cell types might have been co-purified (e.g., as doublets) with eGFP+ endothelial cells during FACS and may not themselves be eGFP+. Pericyte-endothelial doublets are common in scRNA-seq given that these cell types are closely and tightly associated. Additionally, tight association (e.g., via peg-socket junctions) can cause fragments of endothelial cells to be retained on pericytes (and vice-versa) during dissociation. Finally, it is possible that after stroke or during the dissociation process, endothelial cells lyse and release eGFP that could be taken up by other cell types. All of these scenarios could lead to purification of cells that were not derived (transdifferentiated) from endothelial cells. Authors note that the proportion of pericytes increased in the stroke groups, but it does not appear this experiment was replicated and thus this conclusion is not supported by statistical analysis. The results of pseudotime and trajectory analyses rely on the foundation that the pericytes in this dataset are endothelial-derived, which, as discussed above, has not been rigorously demonstrated.

(2) I have the same concern regarding inadvertent purification of cells that were not derived from endothelial cells in the context of the bulk RNA-seq experiment (Fig. S4), especially given the sample-to-sample variability in gene expression in the RP34D, eGFP+ non-ECs group (e.g., only 2/5 samples are enriched for mesenchymal transcription factor Tbx18, only 1/5 samples are enriched for mural cell TF Heyl). If the sorted eGFP+ non-ECs were pericytes, I would expect a strong and consistent pericyte-like gene expression profile.

(3) Authors use immunohistochemistry to understand localization, morphology, and marker expression of eGFP+ cells in situ. The representative "E-pericytes" shown in Fig. 3A-D are not associated with blood vessels, and the authors' quantification also shows that the majority of such cells are not vessel-associated ("avascular"). By definition, pericytes are a component of blood vessels and are embedded within the vascular basement membrane. Thus, concluding that these cells are pericytes ("E-pericytes") may be erroneous.

(4) CD13 flow cytometry and immunohistochemistry are used extensively to identify pericytes. In the context of several complementary lineage tracing strategies noted in Strength #2, CD13 immunohistochemistry is the only marker used to identify putative pericytes (Fig. S3J-M). In stroke, CD13 is not specific to pericytes; dendritic cells and other monocyte-derived cells express CD13 (Anpep) in mouse brain after stroke (PMID: 38177281, https://anratherlab.shinyapps.io/strokevis/).

(5) Authors conclude that "EC-specific overexpression of the Tgfbr2 protein by a virus (Tgfbr2) decreases Evans blue leakage, promotes CBF recovery, alleviates neurological deficits and facilitates spontaneous behavioral recovery after stroke by increasing the number of E-pericytes." All data in Fig. 10, however, compare endothelial Tgfbr2 overexpression to a DsRed overexpression control. There is no group in which Tgfbr2 is overexpressed but "E-pericytes" are eliminated with DTA (this is done in Fig. 9B, but this experiment lacks the Tgfbr2 overexpression-only control). Thus, the observed functional outcomes cannot be ascribed to "E-pericytes"; it remains possible that endothelial Tgfbr2 overexpression affects EB leakage, CBF, and behavior through alternative mechanisms.

In response to this comment, authors wrote: "in Figures 9A-B, we observed no significant difference in Evans blue leakage between the Tgfbr2 overexpression group and the Tgfbr2 overexpression + DTA group (P=0.8153), this suggests that the impact of Tgfbr2 overexpression on the blood-brain barrier (BBB) is primarily attributed from the E-pericytes generated by Tgfbr2 expression."

I do not see data from a Tgfbr2 overexpression-only group in Fig. 9B. Further, I do not understand authors' logic: If the mechanism by which EC Tgfbr2 overexpression acts to reduce BBB leakage is by increasing the number of "E-pericytes," depleting "E-pericytes" with DTA in this context should increase BBB leakage.

(6) Single-cell and bulk RNA-seq data are not available in a public repository (such as GEO). Depositing these data would facilitate their independent reevaluation and reuse.

In response to this comment, authors indicated they submitted data to GEO, but did not provide an accession number.

---

## [Referee Report · Reviewer #3 (Public review)]

Summary:

The data and experiments presented in that study convincingly show that a subpopulation of endothelial cells undergo transformation into pericyte-like cells after stroke in mice. These so-called "E-pericytes" are protective and might present a new target for stroke recovery. The authors used a huge battery of different techniques and modified signaling pathways and cellular interactions using several genetic and pharmacological tools to show that TGFbeta and EndoMT are causes of this transformation.

Strengths:

The amount of different genetic and pharmacological approaches in combination with sophisticated techniques such as single-cell RNAseq is impressive and convincing. The results support their conclusions and the authors achieved their aims. The findings will strongly impact the field of cerebrovascular recovery after stroke and might open up new therapeutic targets.

Weaknesses:

In addition to improving the written and graphical presentation of the results, there is only one point I would like to see clarified: the inclusion of additional experiments, even if they have already been performed but are not applicable due to methodological difficulties regarding the role of Procr+ cells. Negative results also help the scientific community avoid unnecessary experiments and advance understanding.

---

## [Author Response]

The following is the authors’ response to the original reviews

**Public Reviews:**

**Reviewer #1 (Public review):**
Summary:Using lineage tracing and single-cell RNA sequencing, Li et al. reported brain ECs can differentiate into pericytes after stroke. This finding is novel and important to the field.Strengths:Detailed characterization of each time point and genetic manipulation of genes for study role of ECs and E-pericyte.Weaknesses:Genetic evidence for lineage tracing of ECs and E-pericytes requires more convincing data that includes staining, FACS, and scRNA-seq analysis.

We appreciate the reviewer’s recommendation to explore more convincing data, including staining, FACS, and scRNA-seq analysis. We initially employed traditional lineage tracing methods to demonstrate that endothelial cells can transform into pericytes after stroke. We utilized Cdh5CreERT2;Ai47 mice, Tie2-Dre;Mfsd2aCreER;Ai47 mice, and AAV-BI30 virus-infected Ai47 mice. However, in our validation of the transformed cells as pericytes, there are limitations to our results. While three pericyte markers (CD13, NG2, and PDGFRβ) were used in Cdh5CreERT2;Ai47 mice, only one marker (CD13) was applied in Tie2Dre; Mfsd2aCreER;Ai47 and AAV-BI30 virus-infected Ai47 mice. This is insufficient, and the other two pericyte markers (NG2 and PDGFRβ) need to be verified in these models.

At scRNA-seq, although we observed an increased proportion of pericyte/EGFP^+^ cells after stroke, we did not rule out potential contamination by pericyte cells, nor did we include sufficient replicates. To address these issues, we can explore additional methods for analyzing scRNA-seq data, increasing sample replicates, and eliminating pericyte contamination using advanced algorithms. Furthermore, we can use chimeric-related mutations to compare normal endothelial cells, normal pericytes, endothelial-derived pericytes (E-pericytes), and intermediate fibroblast-like cells at the DNA level. This approach will help identify and trace chimeric-related mutations across different cell types and developmental stages. Finally, we can track the entire process of endothelial cell transformation into pericytes using two-photon imaging in vivo.

**Reviewer #2 (Public review):**
Summary:In this manuscript, Li and colleagues study the fate of endothelial cells in a mouse model of ischemic stroke. Using genetic lineage tracing approaches, they found that endothelial cells give rise to non-endothelial cells, which they term "E-pericytes." They further show that depleting these cells exacerbates blood-brain barrier leakage and worsens functional recovery. The authors also provide evidence that endothelial-to-mesenchymal transition, myeloid cell-derived TGFβ1, and endothelial TGFβRII are involved in this process. These are potentially interesting findings, however, the experimental evidence that endothelial cells undergo transdifferentiation to non-endothelial cells is weak, as is the evidence that these cells are pericytes. Addressing this foundational weakness will facilitate the interpretation of the other findings.Strengths:(1) The authors address an important question about blood vessel function and plasticity in the context of stroke.(2) The authors use a variety of genetic approaches to understand cell fate in the context of stroke. Particularly commendable is the use of several complementary lineage tracing strategies, including an intersectional strategy requiring both endothelial Cre activity and subsequent mural cell NG2 promoter activity.(3) The authors address upstream cellular and molecular mechanisms, including roles for myeloid-derived TGFβ.Weaknesses:(1) The authors use Cdh5-CreERT2; Ai47 mice to permanently label endothelial cells and their progeny with eGFP. They then isolate eGFP^+^ cells from control and MCAO RP7D and RP34D brains, and use single-cell RNA-seq to identify the resulting cell types. Theoretically, all eGFP^+^ cells should be endothelial cells or their progeny. This is a very powerful and well-conceived experiment. The authors use the presence of a pericyte cluster as evidence that endothelial-to-pericyte transdifferentiation occurs. However, pericytes are also present in the scRNA-seq data from sham mice, as are several other cell types such as fibroblasts and microglia. This suggests that pericytes and these other cell types might have been co-purified (e.g., as doublets) with eGFP^+^ endothelial cells during FACS and may not themselves be eGFP^+^. Pericyte-endothelial doublets are common in scRNA-seq given that these cell types are closely and tightly associated. Additionally, tight association (e.g., via peg-socket junctions) can cause fragments of endothelial cells to be retained on pericytes (and vice-versa) during dissociation. Finally, it is possible that after stroke or during the dissociation process, endothelial cells lyse and release eGFP that could be taken up by other cell types. All of these scenarios could lead to the purification of cells that were not derived (transdifferentiated) from endothelial cells. The authors note that the proportion of pericytes increased in the stroke groups, but it does not appear this experiment was replicated and thus this conclusion is not supported by statistical analysis. The results of pseudotime and trajectory analyses rely on the foundation that the pericytes in this dataset are endothelial-derived, which, as discussed above, has not been rigorously demonstrated.

Thank you for your thoughtful comment.

Indeed, we face the challenge of obtaining pure cells. As the reviewer has pointed out, several factors may contribute to cell contamination. For instance, the meninges of adult mice are difficult to remove completely, which may lead to fibroblast contamination. Although Cdh5CreERT2 can specifically label endothelial cells in the normal brain parenchyma, there may still be very few unspecific cells in certain brain regions, such as the choroid plexus and periventricular areas, resulting in the presence of ependymal cells. To address these issues, we can improve our methodology by carefully removing the meninges, choroid plexus, and periventricular cells during sample preparation. Additionally, we need to increase the N of the transcriptome samples to enhance the reliability of our data.

(2) I have the same concern regarding the inadvertent purification of cells that were not derived from endothelial cells in the context of the bulk RNA-seq experiment (Figure S4), especially given the sample-to-sample variability in gene expression in the RP34D, eGFP^+^ non-ECs-group (e.g., only 2/5 samples are enriched for mesenchymal transcription factor Tbx18, only 1/5 samples are enriched for mural cell TF Heyl). If the sorted eGFP^+^ non-ECs were pericytes, I would expect a strong and consistent pericyte-like gene expression profile.

This is an interesting question.

Indeed, significant differences were observed in the expression of pericyte-related transcriptional profiles within the eGFP^+^ non-ECs group. For instance, transcription factors such as Hic1 and Fosl1 were nearly absent in the eGFP^+^ non-ECs group. We propose several potential explanations for these observations:

(1) The sorted eGFP^+^ non-ECs group may contain other cell types, leading to contamination.

(2) The eGFP^+^ non-ECs group may not uniformly express all pericyte-related transcriptional profiles.

(3) The temporal dynamics of transcription factor expression (i.e., different factors being expressed at different stages) could contribute to the observed variability.

(4) The heterogeneity in the timing of endothelial-to-pericyte transformation (i.e., some cells have already transformed into pericytes while others are in the process of transformation at the early stage) may result in significant differences in transcriptional profiles.

(3) The authors use immunohistochemistry to understand localization, morphology, and marker expression of eGFP^+^ cells in situ. The representative "E-pericytes" shown in Figure 3A-D are not associated with blood vessels, and the authors' quantification also shows that the majority of such cells are not vessel-associated ("avascular"). By definition, pericytes are a component of blood vessels and are embedded within the vascular basement membrane. Thus, concluding that these cells are pericytes ("E-pericytes") may be erroneous.

Yes, we found that 72.2% of E-pericytes were free and not associated with blood vessels. Normally, pericytes surround blood vessels and connect to endothelial cells. However, in certain diseases, such as Alzheimer's disease, stroke, and diabetic encephalopathy, pericytes can detach from blood vessels. In our stroke model, we observed that pericytes detach from blood vessels. This phenomenon can be explained by two possible scenarios:

(1) After endothelial cells transform into E-pericytes, the E-pericytes detach from blood vessels due to the pathological environment following stroke.

(2) After stroke, blood vessel function is impaired, leading to vascular degeneration. Endothelial cells shed from the blood vessels and subsequently transform into E-pericytes.

Therefore, preventing pericyte detachment from blood vessels after stroke represents an important scientific challenge.

(4) CD13 flow cytometry and immunohistochemistry are used extensively to identify pericytes. In the context of several complementary lineage tracing strategies noted in Strength #2, CD13 immunohistochemistry is the only marker used to identify putative pericytes (Figure S3J-M). In stroke, CD13 is not specific to pericytes; dendritic cells and other monocyte-derived cells express CD13 (Anpep) in mouse brain after stroke (PMID: 38177281, https://anratherlab.shinyapps.io/strokevis/).

We thank the reviewer for their valuable input. In the context of stroke, CD13 is not specific to pericytes. Additionally, pericytes lack a single specific marker; instead, their identity is determined by a combination of multiple markers. To more convincingly validate the identity of pericytes, it is necessary to incorporate additional pericyte markers alongside several complementary lineage tracing strategies.

(5) The authors conclude that "EC-specific overexpression of the Tgfbr2 protein by a virus (Tgfbr2) decreases Evans blue leakage, promotes CBF recovery, alleviates neurological deficits and facilitates spontaneous behavioral recovery after stroke by increasing the number of E-pericytes." All data in Figure 10, however, compare endothelial Tgfbr2 overexpression to a DsRed overexpression control. There is no group in which Tgfbr2 is overexpressed but "E-pericytes" are eliminated with DTA (this is done in Figure 9B, but this experiment lacks the Tgfbr2 overexpression-only control). Thus, the observed functional outcomes cannot be ascribed to "E-pericytes"; it remains possible that endothelial Tgfbr2 overexpression affects EB leakage, CBF, and behavior through alternative mechanisms.

We thank the reviewer for their valuable comment. Although in Figures 9A-B, we observed no significant difference in Evans blue leakage between the Tgfbr2 overexpression group and the Tgfbr2 overexpression + DTA group (P=0.8153), this suggests that the impact of Tgfbr2 overexpression on the blood-brain barrier (BBB) is primarily attributed from the E-pericytes generated by Tgfbr2 expression. Furthermore, in Figure 10A, the inclusion of the Tgfbr2 overexpression + DTA group would provide stronger evidence that the effects of Tgfbr2 overexpression on the BBB and neurobehavioral outcomes are mainly due to the E-pericytes derived from Tgfbr2 expression.

(6) Single-cell and bulk RNA-seq data are not available in a public repository (such as GEO). Depositing these data would facilitate their independent reevaluation and reuse.

Thank you for the suggestion and we have uploaded Single-cell and bulk RNA-seq data (The assignment of GEO number is pending).

**Reviewer #3 (Public review):**
Summary:The data and experiments presented in that study convincingly show that a subpopulation of endothelial cells undergo transformation into pericyte-like cells after stroke in mice. These so-called "E-pericytes" are protective and might present a new target for stroke recovery. The authors used a huge battery of different techniques and modified signaling pathways and cellular interactions using several genetic and pharmacological tools to show that TGFbeta and EndoMT are causes of this transformation.Strengths:The amount of different genetic and pharmacological approaches in combination with sophisticated techniques such as single-cell RNAseq is impressive and convincing. The results support their conclusions and the authors achieved their aims. The findings will strongly impact the field of cerebrovascular recovery after stroke and might open up new therapeutic targets.Weaknesses:The written and graphic presentation of the findings needs substantial improvement. Language editing is strongly recommended (there are a lot of spelling and grammatical errors in the text and illustrations, including legends).

Thank you for raising this important point and we will place greater emphasis on the written and graphic presentation of the findings.

**Recommendations for the authors:**

**Reviewer #1 (Recommendations for the authors):**
In this study, Li et al. reported that endothelial cells in the brain can differentiate into pericytes to promote the restoration of blood-brain barrier (BBB) function after stroke. Understanding the mechanisms underlying BBB restoration post-stroke is crucial to the field. Using lineage tracing, RNA sequencing (RNA-seq), and immunostaining, Li et al. detected the transdifferentiation of endothelial cells (ECs) into E-pericytes in the middle cerebral artery occlusion (MCAO) model. The specific knockout of Tgfbr2 in ECs reduced the number of E-pericytes, exacerbated BBB leakage, and worsened neurological deficits. This observation of EC to pericyte differentiation is novel; however, the conclusions at this stage are not fully supported by the evidence provided.(1) The authors claimed, based on the EdU assay, that 12.9% of pericytes present at RP34D originated from self-proliferation, while the origin of the remaining 27.6% of new pericytes remains unclear. This raises concerns, as the EdU assay is not 100% efficient in detecting all proliferating cells. If EdU^+^ ECs account for fewer than 10% of all ECs, it follows that other EdU-ECs must have alternative origins.

That is an interesting question. To address this issue, we need to consider the following aspects:

(1) The EdU assay is not 100% efficient in detecting all proliferating cells, which means that the actual proportion of proliferating pericytes may be higher than 12.9%, while the proportion of pericytes from other sources may be lower than 27.6% (as determined by FACS). This is consistent with the observation in Figure 3H (immunofluorescence analysis), where EGFP^+^ pericytes accounted for only 24.5% of all pericytes.

(2) The dose of EdU administered in our study was relatively high (200 mg/kg, intraperitoneal injection, daily), which may increase the efficiency of EdU labeling.

(3) When EdU^+^ endothelial cells (ECs) constitute less than 10% of all ECs, it does suggest that EdU-ECs could be a source of pericytes. However, at least EdU^+^ ECs cannot transform into pericytes, as we did not detect any EdU^+^EGFP^+^ pericytes.

(2) The reference for Cdh5CreERT2 is cited as 25, which is a review article published in ATVB. This review lists many different drivers, and the specific Cdh5CreERT2 line used in this study is not identified. This specificity is critical for accurate lineage tracing of ECs.

Although the review I mentioned did not address this, the specificity of Cdh5CreERT2 in the brain has been demonstrated in other studies (Boyé K, et al. Nat Commun. 2022 Mar 4;13(1):1169; Patel A, et al. Proc Natl Acad Sci U S A. 2024 Dec 3;121(49):e2322124121). We have further confirmed that Cdh5CreERT2 specifically labels endothelial cells in the brain parenchyma (Figure S1). Additionally, we found nonspecific labeling in the blood (less than 1% CD45+ blood cells, primarily myeloid cells) and meninges outside the brain parenchyma. We ruled out nonspecific transdifferentiation labeling in the blood through bone marrow reconstitution experiments and in the meninges using in vivo two-photon imaging (results not shown).

(3) The scRNA-seq data should include GFP signals to track the increasing number of pericytes from early to late stages post-injury. This is the only independent method from staining to verify that the pericytes are indeed derived from GFP^+^ ECs after brain injury. Sham samples should be utilized as strict side-by-side controls.

This is a valuable suggestion. We observed that, despite being positive for EGFP protein, only 50% of the sorted cells expressed the EGFP gene at the transcriptome level. This phenomenon has also been reported in other studies (Rodor J,et al a. Cardiovasc Res. 2022 Aug 24;118(11):2519-2534.). For these reasons, we did not rely on GFP signals to track the increase in pericyte numbers from early to late stages post-injury.

(4) Since Ai47 is employed, there are three different variants of green fluorescent proteins, including ZsGreen, which may result in signals being spotted in the staining. The GFP signal detected could also represent dead cells that have lost CD31 expression.

The detected GFP signal could also originate from dead cells that have lost CD31 expression, which is a plausible explanation. As shown in Figure 3I, EGFP^+^ non-ECs peak at RP14D and then decline, suggesting that some EGFP^+^ non-ECs either die or revert to endothelial cells (ECs). Therefore, it cannot be ruled out that we captured some dead EGFP^+^ non-ECs; however, as indicated in Figure 3I, this proportion is likely less than 25%. Additionally, pericytes are prone to death in ischemic and hypoxic environments (Figure 1A), which explains why some of the transformed EGFP^+^ non-ECs may die. Nevertheless, at RP514D, we can still detect EGFP^+^ non-ECs, indicating that a subset of these cells can survive for an extended period (Figure S3F).

(5) The quality of the staining images is not convincing, as some non-ECs and ECs are in close proximity, leading to potential artifacts in signal interpretation. The reviewer cannot rely solely on single staining techniques to be convinced of EC differentiation into pericytes. Although it has been reported that ECs can differentiate into pericytes during development, this phenomenon in the adult brain is surprising; thus, more rigorous evidence with strong lineage tracing data should be provided through multiple measurements.

Why some non-ECs and ECs are located nearby:

(1) Non-ECs exhibit characteristics of pericytes, which are typically adjacent to ECs.

(2) Could this proximity lead to potential artifacts in signal interpretation? We believe this is unlikely, as we also observed a significant number of non-ECs located far from ECs on blood vessels (Figure 3A-B, Figure S3M).

(3) Three pericyte markers (CD13, NG2, and PDGFRβ) were also used to verify the transformed cells, while the three pericyte markers were not expressed in normal endothelial cells.

(6) FACS (Fluorescence-activated cell sorting) should be employed to quantitatively assess the contribution of GFP^+^ ECs to pericytes at each stage after injury, compared to sham controls.

Yes, if the contribution of GFP^+^ ECs to pericytes could be assessed at each time point, the role of E-pericytes in the pericyte pool could be better explained, and the proportion of E-pericytes would become more prominent. In Figure 3, we did not use FACS to evaluate the contribution of GFP^+^ ECs to pericytes at each stage post-injury. Instead, we only assessed the ratio of EGFP^+^ non-ECs to all EGFP^+^ cells. However, we did verify the contribution of GFP^+^ ECs (E-pericytes) to pericytes at RP34D using FACS (CD13+ DsRed/CD13 = 25.6%, Figure 4C). This ratio is consistent with the immunofluorescence data (Figure 3H).

(7) In Tie2Dre;Mfsd2aCrexER;Ai47 mice, ECs in the brain are specifically labeled, indicating that ECs could give rise to CD13+ EGFP^+^ non-ECs at RP34D (Figure S3L). However, the GFP signal for Ai47 is not homogeneous, displaying many spotted patterns. Using tdTomato as an alternative for detection could enhance clarity.

We repeated the experiment using tdTomato as the reporter gene in mice and observed results consistent with those obtained using Ai47 as the reporter gene. For consistency, all results presented are based on Ai47. Regarding the spotted patterns observed with Ai47, this phenomenon can be attributed to the relatively low laser intensity (2%). Higher laser intensity would cause overexposure of EGFP^+^ ECs. To address the issue of spotted patterns in Ai47 imaging, we can improve the visualization of complete cell morphology (as shown in Figure S3M) by increasing the gain value, which enhances the background signal.

(8) The data concerning the genetic ablation of pericytes lacks specificity. There is insufficient evidence to support that DTA is specifically expressed in E-pericytes. The authors should utilize DTR (Diphtheria Toxin Receptor) and confirm that DTR expression is restricted to pericytes derived from GFP^+^ ECs. Treatment with diphtheria toxin, but not PBS as a control, should specifically ablate these E-pericytes without affecting any other GFP-pericytes in the brain following injury.

We did not verify that DTA expression was restricted to E-pericytes. To ensure that DTA is only expressed in converted E-pericytes, we employed two strategies:

(1) Specific Targeting of Endothelial Cells: We used the AAV-BI30 virus to specifically infect endothelial cells. Although not 100% exclusive, 98.5% of the expression occurred in endothelial cells, with minimal infection in neurons and microglia. Additionally, we combined this with Cdh5CreERT2 to control the DIO action in the virus. This means that only endothelial cells expressing both Cdh5CreERT2 and infected with AAV-BI30 could undergo cell fate changes and transform into pericytes, subsequently expressing markers such as NG2 and driving DTA expression in E-pericytes (Figure 4A).

(2) Validation of DTA Expression: To prevent off-target expression of DTA in other cell types, we plan to verify DTA protein expression using specific antibodies to confirm whether DTA is expressed in unintended cells. Alternatively, as suggested, we could utilize the Diphtheria Toxin Receptor (DTR) system. By ensuring that DTR expression is restricted to pericytes derived from GFP^+^ ECs, treatment with diphtheria toxin would specifically ablate these E-pericytes without affecting other GFP- pericytes in the brain post-injury.

(9) There is currently no convincing genetic data demonstrating that Tgfb signaling overexpression or deletion modulates the transdifferentiation of ECs to pericytes.

Yes, this is an important consideration. Although we knocked out the TGFβ receptor in endothelial cells (ECs) and observed a reduction in the formation of E-pericytes (Figure 6D and 6G), it would be more informative to specifically knockout the Tgfb gene in myeloid cells or monocyte-macrophage lineages to determine whether these cells are the primary source of TGFβ driving endothelial cell transformation. Additionally, injecting TGFβ protein directly into the brains of mice could help explore whether exogenous TGFβ promotes the formation of E-pericytes.

**Reviewer #2 (Recommendations for the authors):**
(1) Figure 1D, there does not appear to be a clear PDGFRβ-positive population. In this case, it is necessary to include the negative control that served as the basis for drawing the positive gate.

Author response image 1 below show the negative control for CD31 and PDGFRβ.

**Author response image 1. sa4fig1:** 

(2) Figures 3A-D, Figures S3J-M, the authors statistically compare % negative to % positive. It appears % negative = 100% - % positive. If this is the case, these groups are not independent and should not be statistically compared.

This is a very important point, and such a comparison is not appropriate. The statistical comparison mentioned above has now been removed.

(3) Figure 4B, in addition to the cells indicated with arrows, there is a substantial additional DsRed+ signal of similar intensity in this image. It would be helpful to show a negative control.

Author response image 2 below show the contralateral and ipsilateral, respectively. In the contralateral, DsRed has few signals, no complete cell morphology, and is separated from the Hoechst+ nucleus. in the ipsilateral, DsRed signals are strong, have intact cell morphology, and are tightly bound to the Hoechst+ nucleus. In the ipsilateral, some DsRed signals may come from dying cells.

**Author response image 2. sa4fig2:** 

(4) Figure 6G, the y-axis title is "E-pericytes/all EGFP^+^ cells (%)" but the y-axis scale goes from 0 to 900. Is this an error?

Thank you. We want to calculate the number of pericytes per unit area, it should be E-pericyte/mm2.

(5) Figure 9B, in the representative images, the 6th group is labeled "Tgfb2 + DTA" but in the plot below, the 6th group is labeled Tgfbr2 + DsRed. Which is correct?

Thank you. The "Tgfb2 + DTA" is right. We have changed it to "Tgfb2 + DTA" in the 6th group, Figure 9B.

(6) Figure S1I, error bars and/or individual data points should be shown.

The purpose of this diagram is to demonstrate the number of mice in which EGFP^+^ cells are 100% co-labeled with endothelial markers (CD31, ERG, GLUT1, and VE-Cadherin), as EGFP^+^ cells are exclusively found in endothelial cells within the brain parenchyma. Additionally, the diagram illustrates the number of mice in which EGFP^+^ cells show no co-labeling (0%) with mural cell markers (CD13, PDGFRβ, α-SMA, and NG2), as EGFP^+^ cells are not present in mural cells within the brain parenchyma.

(7) The authors write: "When Tgfbr2 was overexpressed and DTA was expressed specifically in the same ECs, DTA prevented the EC-specific overexpression of the Tgfbr2 gene and increased the proportion of E-pericytes.". The authors' strategy for DTA expression involves the NG2 promoter, which, in principle, is not active in ECs. Thus how can DTA be "expressed specifically in the same ECs" and how can DTA "prevent EC-specific overexpression" of Tgfbr2?

Our purpose is not clearly expressed. The statement should be revised to: “When Tgfbr2 was overexpressed to increase E-pericytes and DTA was expressed in transformed cells to deplete E-pericytes, we found that there was no significant change in the number of E-pericytes in the Tgfbr2 + DTA group compared with the DTA group.”

(8) The interpretation of Evans blue leakage as "low molecular weight" leakage should be revised since Evans blue binds serum albumin and thus it is the molecular weight of this complex (~67 kDa) that is relevant.

We agree with the reviewer. Yes, it should not be stated that Evans blue is low molecular weight, as it binds to serum albumin to form complexes. The text has been revised to: “Interestingly, no obvious leakage of dextran-rhodamine B (~70 kDa) (Figure S8C) or Texas Red (~71 kDa) was detected (Figure S8D). However, the elimination of E-pericytes allowed evans blue and trypan blue to cross the blood-brain barrier (BBB).”

(9) It is critical that the sequencing data be made available through a public repository (such as GEO).

Thank you. Now we've uploaded it to GEO.

(10) It would be extremely helpful if the authors would make their viral plasmids available through a public repository (such as Addgene).

Thank you. Now we've uploaded it to Addgene (The assignment of Addgene number is pending).

**Reviewer #3 (Recommendations for the authors):**
(1) The distribution and expression of pericytic and fibroblast markers at different time points after stroke is confusing while reading the manuscript, e.g., vimentin is not expressed on day 34 but on day 8, whereas CD13 is expressed on day 34 but not on day 8, if I understood the text correctly. To make it easier to follow, the authors could add a label of the day after stroke to each of the subfigures which show images and co-expression of different markers (e.g. Figures 3 and S3).

Below are the expressions of different specific markers in each cell.

“√” stand for positive, “×” stand for positive

**Author response table 1. sa4table1:** 

Markers	ECs	Fibroblast-like cell (RP8D)	Pericyte (RP34D)
CD31	✓	xx	xx
GLUT1	✓	xx	xx
ERG	✓	xx	xx
VE-Cadherin	✓	xx	xx
PDGFRa	xx	✓	xx
PDGFR beta	xx	✓	sqrt()
Vimentin	xx	✓	xx
CD13	xx	xx	✓
NG2	xx	xx	sqrt()

(2) The authors need to check the N numbers again, e.g., Figure S3L: 4 dots per group are shown in the graph but an N of 3 is mentioned in the legend.

Thank you for raising this important point. N=4 has been corrected in the legend of Figure 3S. We also checked other N numbers.

(3) Labelling of graphs should be consistent (e.g., S4C: "I-ECs" vs. S4F: "Ipsi-ECs") and correct (e.g., "DsRed" instead of "DeRed" in Figure 4B).

Yes, we need a uniform name with "Ipsi-ECs" and "DsRed". Thank you.

(4) Figure 4: In the text, the injection is described to be done on day 34 whereas in Figure 4A the injections are described to take place before MCAO, please clarify. Does day 34 mean 34 days after injection or after MCAO (as in the former experiments)?

In the text, the sentence, “Then we used AAV2/9-BI30-NG2 promoter-DIO-DTA (DTA) to deplete E-pericytes at RP34D (Figure 4D),” could be misinterpreted as suggesting that the virus was injected at RP34D. To avoid confusion, it has been revised to: “We used the AAV2/9-BI30-NG2 promoter-DIO-DTA (DTA) virus, which was injected before MCAO (Figure 4A), to deplete E-pericytes (Figure 4D).” Yes, day 34 means 34 days after injection or after MCAO and we unify to 34 days.

(5) Some images are too dark to recognize clear structures and prove the findings (e.g., Figure S6B).

Thank you for raising this important point.

(6) There is no Figure S8D (as mentioned in the text).

Thank you for raising this important point. This problem has been corrected.

(7) Figure S9: the text only states, that Tgfbr2 overexpression increases CBF recovery and effective perfusion. Also with the legend, it is not clear what was done and measured, especially in Figure S9B - what do the graphs show? Also, the y-axis labeling is missing for the traces.

In Figure S9A, we assessed changes in blood flow using laser speckle imaging. Laser speckle imaging relies on random interference patterns formed by scattered light when a laser strikes tissue. Moving red blood cells alter the contrast of the speckle pattern: faster blood flow results in quicker speckle changes and lower contrast, while slower blood flow leads to slower speckle changes and higher contrast. By analyzing these changes in speckle contrast, blood flow dynamics can be evaluated in real-time and non-invasively.

In Figure S9B, we measured blood flow changes using Laser Doppler flowmetry. When a laser interacts with flowing blood, the moving red blood cells scatter the light, causing a frequency shift (Doppler shift). Faster blood flow results in a greater frequency shift, while slower blood flow leads to a smaller frequency shift. By detecting the frequency shift of the scattered light, blood flow velocity and changes can be measured in real time and non-invasively. In Laser Doppler Flowmetry (LDF), the unit of the vertical axis is typically Perfusion Units (PU). PU is a relative unit used to represent changes in blood flow rather than absolute blood flow velocity. These methods have now been further explained in the diagram.

(8) Which regions of the brain were used to take images (e.g., to count neurons)?

We captured images and quantified neurons in the cortex and striatum of the brain. Our statistical analysis further demonstrated that, at RP34D, the presence of E-pericytes in the brain does not exhibit region-specificity. Instead, the formation of E-pericytes is driven by TGFβ1, which is regulated by immune cells. Ultimately, the distribution and activity of these immune cells are influenced by the severity of ischemia and hypoxia.

(9) The sentence "Protein C receptor-expressing (Procr+) ECs could give rise to de novo formation of ECs and pericytes in the mammary gland13." is repeated almost identically in three different places in the text. However, whether Procr+ cells are involved in the described transdifferentiation or whether "E-pericytes" do express the protein C receptor is not shown and needs additional investigation.

The reason for referencing this literature is to highlight that endothelial cells (ECs) during breast development can give rise to pericytes, which serves as background knowledge supporting our research. To further explore this phenomenon in brain, we used ProcrCreERT2;Ai47 mice subjected to MCAO (middle cerebral artery occlusion) to investigate whether Procr+ ECs could transform into pericytes, similar to what occurs in mammary glands. However, since ProcrCreERT2 labels not only ECs but also pericytes in the brain, the results did not achieve our goal and were therefore not included in the study.